



# Coastal and orographic effects on extreme precipitation revealed by weather radar observations

Francesco Marra[1], Moshe Armon[2], Efrat Morin[2]

[1]National Research Council of Italy, Institute of Atmospheric Sciences and Climate (CNR-ISAC), Bologna, 40129, Italy

[2]The Fredy & Nadine Herrmann Institute of Earth Sciences, the Hebrew University of Jerusalem, Jerusalem, Israel

*Correspondence to*: F. Marra (f.marra@isac.cnr.it), Via Gobetti, 101, 40129, Bologna, Italy, *Phone*: +39 0516398084

**Abstract.** The yearly exceedance probability of extreme precipitation of multiple durations is crucial for infrastructure design, risk management and policymaking. Local extremes emerge from the interaction of weather systems with local terrain features such as coastlines and orography, however multi-duration extremes do not follow exactly the patterns of cumulative precipitation and are still not well understood. High-resolution information from weather radars could help us better quantifying their patterns, but traditional extreme-value analyses based on radar records were found too inaccurate for quantifying the extreme intensities for impact studies. Here, we propose a novel methodology for extreme precipitation frequency analysis based on relatively short weather radar records, and we use it to investigate coastal and orographic effects on extreme precipitation of durations between 10 minutes and 24 hours. Combining 11 years of radar data with 10-minute rain gauge data in the southeastern Mediterranean, we obtain estimates of the 1 in 100 years intensities with ~22% standard error, which is lower than those obtained using traditional approaches on rain gauge data. We identify three distinct regimes, which respond differently to coastal and orographic forcing: short durations (~10 minutes), related to peak convective rain rates; hourly durations (~1 hours), related to the yield of individual convective cells; and long durations (~6-24 hours), related to the accumulation of multiple convective cells and to stratiform processes. At short and hourly durations, extreme return levels peak at the coastline, while at longer durations they peak corresponding to the orographic barriers. The distributions tail heaviness is rather uniform above the sea and rapidly changes in presence of orography, with opposing directions at short (decreasing tail heaviness, with a peak at hourly durations) and long (increasing) durations. These distinct effects suggest that short-scale hazards such as urban pluvial floods could be more of concern for the coastal regions, while longer-scale hazards such as flash floods could be more relevant in mountainous areas.

## 1 Introduction

Knowledge of the yearly exceedance probability of extreme precipitation intensities at multiple spatiotemporal scales is crucial for infrastructure design, weather-risk management and policymaking (Chow et al., 1988; Kleindorfer and Kunreuther, 1999). Typically, statistical models are fitted to the observed extremes and used to extrapolate design precipitation intensities corresponding to low yearly exceedance probabilities, which cannot be empirically quantified from observations. For example,



the 100-year return levels are intensities characterized by 1% yearly exceedance probability, and thus exceeded on average once in 100 years (Katz et al., 2002). This task usually requires specific approaches to decrease the stochastic uncertainties characterizing the observation of extremes (e.g., Koutsoyiannis et al. 1988; Buishand, 1991; Burlando and Rosso, 1996) and simplified conceptual models to account for the multiple spatiotemporal scales required by many impact studies (Sivapalan and Blöschl, 1998; Svensson and Jones, 2010).

Local precipitation climatology emerges from the interaction between weather systems and local terrain features. For instance, mountains constrain the flow of air masses inducing vertical motions which affect precipitation (Haiden et al., 1992; Houze et al., 2001; Roe, 2005). Similarly, sea-land interfaces and lakes may alter the precipitation characteristics due to the interaction of synoptic-scale winds with meso-scale features such as sea/land breeze, the frictional-induced convergence associated with deceleration of wind speeds when air is blowing from sea inland, and the contribution of the coastal shape to convergence,

with effects which can propagate far inland (Bergeron, 1949; Neumann, 1951; Bummer et al., 1995; Niziol et al., 1995; Colle and Yuter, 2007; Heiblum et al., 2011; Warner et al., 2012; Li and Carbone, 2015; Minder et al., 2015). While these phenomena are relatively well understood in terms of average precipitation yield, less is known about their impact on extreme precipitation intensities.

Extreme precipitation intensity does not follow exactly the patterns of cumulative precipitation. For example, the typical

orographic enhancement of total precipitation was found reversed for extreme hourly precipitation (the "reverse orographic effect"), meaning that the hourly extremes tend to decrease with elevation (Allamano et al., 2009; Avanzi et al., 2015). The only study available so far on the orographic impact on sub-hourly design precipitation focused on the case of Mediterranean cyclones in the southeastern Mediterranean, and found that orography tends to decrease the tail heaviness of precipitation intensity distribution, with a stronger effect at hourly durations with respect to sub-hourly or multi-hour (Marra et al., 2021b).

As for the sea-land interfaces, extreme precipitation was found to be higher along narrow coastal regions (Mapes et al., 2003; Wu et al. 2009). For the southeastern Mediterranean, rain amounts defining the highest precipitation percentiles at sub-storm durations were found to be higher along the coastline, extending few kilometers inland (Sharon and Kutiel, 1986; Karklinsky and Morin, 2006; Peleg and Morin, 2012; Armon et al., 2020). Nevertheless, with the exception of coarse-scale climatology from satellite-borne sensors, which however did not directly address the issue (Marra et al., 2017; Demirdjian et al., 2018;

Courty et al., 2019), not much is known about the sea-land interactions in terms of extreme return levels, mostly due the obvious lack of in-situ observations over the sea.

The interest in these questions is enhanced by the ongoing climate change, which affects the hydrological cycle in yet unclear manners (Blöschl et al., 2019) and may result in non-linear response to natural hazards triggered by hydroclimatic extremes, such as urban floods (Arnbjerg-Nielsen et al., 2013), flash-floods (Hicks et al., 2005; Amponsah et al., 2018), slope failures

(Marchi et al., 2009; Paranunzio et al., 2019), forest disturbances (Ulbrich et al., 2001; Forzieri et al., 2020), and coastal flooding (Wahl et al., 2015; Bevacqua et al., 2019). An improved understanding of the interaction of weather systems with local terrain features and its effects on design precipitation intensities could help us better representing present and future extremes in impact studies (e.g., Attema and Lenderink, 2014; Peleg et al., 2020). However, information from in-situ



observations is often limited due to the large extension of ungauged regions worldwide (Kidd et al., 2017) and to the sparse
representativeness of rain gauge sampling, particularly important in the case of short duration extremes (Villarini et al., 2018;
Peleg et al., 2018; Marra and Morin, 2018; Lengfeld et al., 2020).

Weather radars monitor regional-scales over both land and sea with high spatial and temporal resolutions, providing fine-scale
information on the spatial gradients of extreme intensities ranging from sub-hourly to sub-daily and daily durations (Collier et
al., 1989; Saltikoff et al., 2019). They represent a unique opportunity to examine these local effects on the statistics of extreme
precipitation, as they cover orographic regions more homogeneously than the typical rain gauge networks, as well as offshore
sea areas up to hundreds of kilometers from the coast (Kidd et al., 2017). The use of radar records for precipitation frequency
analysis currently suffers from important uncertainties, mostly related to measurement errors (Eldardiry et al., 2015; Marra et
al., 2015; McGraw et al., 2019) and to the use of statistical methods inadequate for the characteristics of weather radar records
(Marra et al., 2019a). Most of the efforts so far focused on the reduction of these uncertainties (Overeem et al., 2009; Wright
et al., 2013; Goudenhoofdt et al., 2017), so that climatological analyses were only seldom attempted (Panziera et al., 2016;
Marra et al., 2017).

Recent advances in the statistical description of extreme precipitation showed promising advantages over traditional
approaches in estimating design precipitation intensities in presence of problems typical of radar monitoring, such as short
data records and measurement errors (Marani and Ignaccolo, 2015; Zorzetto et al., 2016; Marra et al., 2018). These novel
methods were tested on satellite data with encouraging results (Zorzetto and Marani, 2019; Zorzetto and Marani, 2020; Hu et
al., 2020; Mekonnen et al., 2021) but, to the best of our knowledge, no weather radar application is available so far, with the
exception of a single storm study case (Rinat et al., 2021).

Here, we exploit these novel statistical methods for precipitation frequency analysis to combine an 11-year weather radar
archive covering the southeastern Mediterranean with data from a long recording rain gauge network. We use this adjusted
dataset to investigate the statistical characteristics of extreme precipitation emerging from the interaction of weather systems
with coastal and orographic features. Specifically, we (a) devise and test a novel methodology to adjust radar-derived statistics
based on in-situ observations, and we use it to (b) investigate the coastal and orographic effects on the statistics of design
extremes at multiple sub-daily durations. Using the new methodology, we find a notable impact of the coast on precipitation
extremes, and characterize differential orographic effects on extreme precipitation over sub-hourly to daily durations.

## 2 Study area and data

### 2.1 Study area

This study focuses on the region covered by the C-band weather radar of the Israeli Meteorological Service (IMS), located at
Bet Dagan, Israel (Fig. 1). The region shows a longitudinally-organized physiography that, moving east from the
Mediterranean Sea, encounters a coastal plain, an orographic barrier (Judean mountains, ~1000 m a.s.l.), a deep rift valley
(Jordan rift valley, ~400 m below sea level), and a second orographic barrier (Jordanian plateau, >1000 m a.s.l.). Weather





systems strongly interact with these rather regular orographic features generating steep gradients in precipitation climatology, which include drops in the mean annual amounts as high as ~500 mm yr$^{-1}$ within a distance of 25 km. Overall, precipitation amounts tend to increase with latitude and elevation and decrease with distance from the coast (Alpert and Shafir, 1989; Goldreich, 1994).


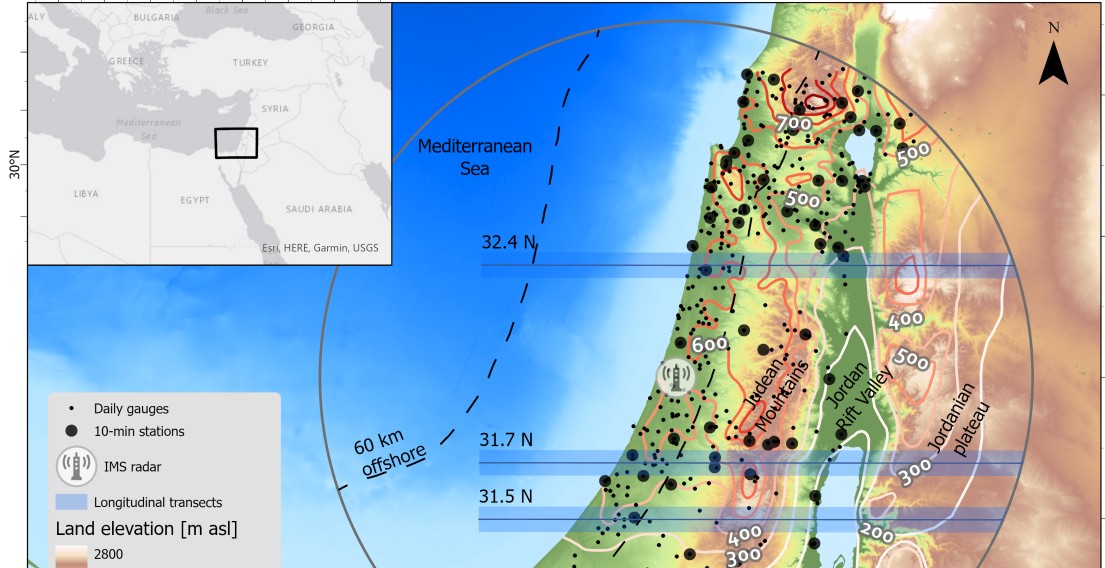

**Figure 1: Map of the study area showing: terrain elevation; location of the weather radar; 140-km range of the instrument; location of the daily rain gauges; location of the 10-minute rain gauges; location of the three transects. Mean annual rainfall is based on 1961-1990 rain gauge data (Enzel et al., 2003).**


Precipitation is mostly brought by (i) Mediterranean cyclones (Ziv et al. 2015; Kushnir et al. 2017), (ii) low-pressure systems extending from the south (Red Sea troughs; de Vries et al., 2013) and (iii) more rarely by conveyor belts of moisture of tropical origin (tropical plumes; Rubin et al., 2007; Tubi and Dayan, 2014). Although these systems are characterized by distinct spatial and temporal scales and all potentially cause extreme precipitation (Marra et al., 2019b), Mediterranean cyclones are the

dominant systems both in terms of number of wet days and total precipitation yield (~75-90% of total amounts), with the exception of the southern and eastern portions of the domain in which Red Sea troughs can contribute almost equally (Belachsen et al., 2017; Armon et al., 2018; Marra et al., 2019b). Mediterranean cyclones typically bring westerly winds from the Mediterranean Sea inland, and thus interact with the coastal and orographic features of the area in a rather systematic way.



Heavy precipitation processes are generally convective and characterized by high spatiotemporal variability, so that rain gauge
monitoring is often insufficient (Yakir and Morin, 2011; Peleg et al., 2013; Marra and Morin, 2018). Climate models project
substantial changes in both the intensity and the occurrence of these systems (Hochman et al., 2018; Zappa et al., 2015), with
complicated effects on compound extremes (Marra et al., 2019b; Marra et al., 2021a).

## 2.2 Rain gauge data

Rain gauge data were provided by the IMS and organized into two archives: (i) daily archive, containing precipitation amounts
measured every day at 6UTC, used for the calibration and validation of the weather radar archive and for the definition of
storms (see below); (ii) 10-minute archive from automatic stations, containing data at 10 min temporal intervals and 0.1 mm
tip resolution, used for the adjustment of radar-derived statistics at multiple temporal scales. All the data were quality-
controlled by the IMS; in addition, stations located in regions with low weather radar data quality (e.g., residual contamination
by ground clutters and blockages) were removed to avoid negative impacts on the adjustments. In total, 437 daily stations and
65 10-minute stations are used in the analyses (Fig. 1). Hydrological years (September 1 to August 31) with more than 10%
missing data were removed to ensure accurate quantification of the precipitation statistics (see Marra et al., 2020).

## 2.3 Weather radar data

A full re-elaboration of the weather radar archive was carried out to obtain a high-quality and homogeneous archive for the
period 2007-2008 to 2017-2018. Physics-based corrections and empirical adjustments are combined in a procedure optimized
for long-term radar archives and high-intensity convective storms (Marra et al., 2014; Marra and Morin, 2015; Marra and
Morin, 2018). Non-precipitating echoes (i.e., ground clutters) are removed using a reparameterization of the filter by Jacobi et
al. (2014) (details in Marra and Morin, 2018). Beam blockage due to orography is corrected (up to 70% power loss, Marra et
al. 2014) using numerical simulations of the beam propagation in a digital elevation model (Pellarin et al., 2002). Non-
orographic blockages are identified from the systematic power loss in long-term (yearly) observations and corrected by
interpolation with nearby non-blocked beams. Beam attenuation in heavy rain is corrected with a modified Hitschfeld and
Bordan (1954) algorithm using an upper limit of 10 dBZ to the correction factor (Marra and Morin, 2015); coefficients for
convective environment are derived from Delrieu et al. (2000). Vertical variations of reflectivity are corrected based on
spatially-averaged profiles observed during two-hour windows using the procedure in Marra and Morin (2015). Reflectivity is
saturated at 59.5 dBZ to avoid contamination by hail. A fixed power law relation in the form $Z = 316 \cdot R^{1.5}$, which was shown
to be adequate for the convective precipitation of the region, is used to compute rain intensity (Morin and Gabella, 2007). Rain
rate at the ground is obtained averaging the two highest intensities along the vertical dimension for elevations up to 5°, and is
converted to 500-meter Cartesian grid. Bias adjustment is performed using the daily rain gauge archive and consists of two
steps: range-dependent adjustment based on yearly accumulations (Morin and Gabella, 2007); event-based mean-field





adjustment (Marra and Morin, 2015), with events defined as consecutive wet days (≥5 stations recording ≥0.1 mm, more details in Sect. 3.1).

Leave-one-out cross-validation of the archive at the daily scale shows a median fractional standard error (FSE, that is the root mean square error normalized over the average rain gauge amount) of 1.07, and a median correlation coefficient of 0.76 (see additional details in Figure S1 in the Supporting Information). These results represent a large improvement over the previous radar archive available for the region (Marra and Morin, 2015), which are attributable to the more recent radar instrument and to the updated elaboration procedures, and which motivate our choice of exploiting this shorter but more accurate archive in our study. The instrument was sometimes turned off so that the archive cannot be considered complete. Despite the evaluation metrics, some issues still remain in the final estimates: ground clutters in the first orographic range east of the radar can appear due to anomalous beam propagation; some blocked beams could be not completely recovered or they were over-filled by the correction, especially west of the instrument (e.g., see Fig S1 and Fig. S2); in eastern portion of the domain, the radar sample volumes are quite high in the atmosphere (due to the presence of the first orographic barrier) and no rain gauge is available for adjustment, leading to potential underestimation due to overshooting of the precipitation. Pixels located within 10 km from the instrument have been removed from the subsequent analyses to avoid the artifacts related to the suppression of side-lobe ground echoes. Underestimation due to range effects is also visible in the northern and southern portions of the domain.

## 3 Methods

### 3.1 Extreme value analysis

Design precipitation intensities are here quantified as return levels corresponding to low yearly exceedance probabilities. We use the framework for sub-daily precipitation frequency analysis based on the concept of ordinary events presented in Marra et al. (2020). Ordinary events are defined as all the independent realizations of a process of interest. Once the intensity distribution describing their tail is known, it is possible to compute an extreme value distribution to quantify intensities associated with rare yearly exceedance probabilities by explicitly considering their yearly occurrence frequency (Zorzetto et al., 2016). We define here the ordinary events' tail as the largest 45% of the events, following the results in Marra et al. (2019b) which showed that values between 45% and 20% provide virtually indistinguishable results. We use a stretched-exponential model (Weibull, as proposed by Wilson and Tuomi, 2005) for their tail, which was shown to be optimal for the region (Marra et al., 2020; Marra et al., 2021b). This means that the largest 45% of the ordinary events is approximated using a cumulative distribution function in the form $F(x; \lambda, \kappa) = 1 - e^{-\left(\frac{x}{\lambda}\right)^{\kappa}}$, where $\lambda$ is a scale parameter which scales all the intensities by a common factor, and $\kappa$ is a shape parameter which affects the distribution tail heaviness as follows: $\kappa = 1$ implies that exceedance probability decreases exponentially with intensity, $\kappa > 1$ implies a faster decrease (light tail) and $\kappa < 1$ a slower decrease (heavy tail).





The computation of extreme return levels relies on the simplified Metastatistical extreme value (SMEV) framework (Marra et

al., 2019; Marra et al., 2020). This approach is well suited for our study case because it is less sensitive than traditional extreme value approaches (i) to measurement errors typical of radar estimates (Marra et al., 2018), and (ii) to the use of short records (Zorzetto et al., 2016; Marra et al., 2018; Hu et al., 2020), and because (iii) it correctly represents the tail of sub-daily precipitation intensities (Marra et al., 2020; Wang et al., 2020). For these reasons, it allows to accurately compute at-site parameters and return levels without the need for spatial homogeneity and/or scaling assumptions, thus allowing to examine

the spatial and temporal patterns directly from observations. The method proceeds as follows:

1. Storms are defined at the regional scale as consecutive wet periods of the same weather type separated by 1-day dry periods. Storms are defined based on the daily rain gauge archive as consecutive wet days ($\geq 5$ stations recording $\geq 0.1$ mm). Weather types are defined according to the classification presented in Marra et al. (2021a), which is a simplification of the semi-objective classification by Alpert et al. (2004). It is worth noting that, although the framework allows to

separately consider different weather types, the use of a unique weather type is not crucial for the accurate estimation of return levels in the study area (see Marra et al. 2019); separate consideration of the 2 weather types is here included to grant compatibility with future studies involving climate projections, for which the distinct use of weather types is crucial (e.g., Marra et al., 2021a).

2. Ordinary events are defined at each station/pixel as maximum intensities observed within each storm using moving

windows (10-min steps) of durations between 10 minutes and 24 hours (10, 20, 30 minutes, 1, 2, 3, 6, 12, 24 hours). Ordinary events are computed at-site from both weather radar estimates (each radar pixel) and 10-minute rain gauges (65 stations). Sensitivity analyses based on 1-minute rain gauge data in the region showed that using 10-minute data to quantify annual maximum intensities may lead to an average ~10% underestimation of the 10-minute maximal intensities, which sharply decrease to <6% for 20-minute and <2% for hourly intensities (Marra and Morin, 2015); as we rely on ordinary

events rather than annual maxima, we expect our analyses to be even less affected by this issue;

3. Parameters $\lambda$ and $\kappa$ of the intensity distribution are estimated, for each duration, by left-censoring the lowest 55% of the ordinary events (that is retaining their weight in probability but censoring their magnitude) and using a least-square regression in Weibull-transformed coordinates (Marani and Ignaccolo, 2015); the parameter n (average yearly number of storms, and thus of ordinary events; Fig S2) is computed as the total number of storms which are locally wet (i.e., $\geq 0.1$

mm storm accumulation, and length of the storm $\geq 30$ minutes) divided by the number of years in the record;

4. Extreme return levels $x$ corresponding to the yearly non-exceedance probability $\zeta$ are computed inverting the SMEV distribution $\zeta(x) = F(x; \lambda, \kappa)^n$.

## 3.2 Application to weather radar record

In order to reduce the impact of the systematic biases and random errors previously found to dominate radar-derived frequency

analyses (e.g. Eldardiry et al., 2015, Marra and Morin, 2015) and to account for the potential missing of storms in our radar





archive, we devised a framework to integrate weather radar and rain gauge information in the SMEV methodology. The idea is to adjust the radar estimates within the SMEV parameter-space. This would allow to fully exploit the quantitative accuracy of rain gauges as well as the spatial information from the weather radar. Our approach relies on two assumptions: (i) rain gauge records are good representations of the local statistics of extreme precipitation; (ii) the periods in which the radar was turned off are independent of the ordinary events intensity (i.e., the intensity distribution of the sampled ordinary events can be considered the same as the intensity distribution of all ordinary events). The adjustment proceeds as follows:

1. The SMEV parameters $(\lambda, \kappa, n)$ for each duration are estimated for all radar pixels and for all the 10-minute rain gauges;

2. In order to account for possible missing storms in the radar archive, the radar-derived parameter n (i.e., the average number of yearly storms, which is the same for all durations) is estimated by dividing the number of storms which are locally wet (i.e., on the pixel) by the fraction of storms (as defined in section 3.1) which are available in the radar record;

3. At each rain gauge $j$, the local multiplicative bias (of the radar pixel above the gauge compared to the gauge) in the three SMEV parameters is calculated as: $\text{bias}_j = \frac{z_j^{\text{radar}}}{z_j^{\text{gauge}}}$, where z is the examined parameter: $\lambda, \kappa$, or $n$;

4. The biases are interpolated using an inverse distance weighted method, which simultaneously considers horizontal and vertical distances to produce maps of parameters bias on the whole radar grid. Inverse distance weighting was chosen because it allows to easily consider vertical distance, an important explanatory variable for weather radar bias (e.g., Andrieu et al. 1997) and SMEV parameters (Marra et al., 2021b; Mekonnen et al., 2021), and it was reported to outperform other interpolation methods in estimating extreme return levels using SMEV (Miniussi and Marra, in review). The distance from a generic rain gauge j is thus computed as: $d_j = [x^2 + y^2 + (W \cdot z)^2]^{\frac{1}{2}}$. Vertical distances are over-weighted by a factor $W = 150$, meaning that 100 m in vertical distance are weighted as much as 15 km in horizontal distance. This factor is roughly two times the typical orographic gradients in the area, and was defined after sensitivity analyses which showed that roughly analogous results are obtained for $100 \lesssim W \lesssim 200$, while significantly different results are obtained for weights $W \simeq 0$ (vertical distance is neglected) and $W \gtrsim 500$ (terrain elevation becomes the dominant predictor). Due to the sparse distribution of 10-minute rain gauges in high terrain elevations (Fig. 1), it was not possible to further optimize this factor. Weights were then computed as the square of the inverse distance $w_j = d_j^{-2}$ and normalized over their sum; at each point we used up to 25 neighboring rain gauges;

5. At each pixel, the three SMEV parameters are adjusted by dividing the radar-derived value by the corresponding interpolated biases;

6. Adjusted return levels are computed by inverting the SMEV distribution.

This empirical procedure implicitly corrects biases due to the spatial mismatch between the rain gauge scale and the radar sampling volume (e.g., see Peleg et al., 2018), such as the ones explicitly addressed in the framework by Zorzetto and Marani (2019), so that the resulting return levels represent the "point" scale of rain gauge measurements.



### 3.3 Validation

Quality of the radar-derived return levels is quantified using a bootstrap procedure in which 50% of the stations are randomly removed from the adjustment and used for validation:

1.    Half (50%) of the stations are randomly selected as validation stations;

2.    The radar statistics are adjusted using the remaining 50% of the stations (see above) and the 100-year return levels are calculated;

3.    Points 1-2 are iterated $10^3$ times and the fractional standard error (FSE, see section 2.3) corresponding to the 100-year return levels is computed.

In order to quantify the quality of the return levels estimated using SMEV on the relatively short weather radar archive (11 years), we compared the FSE with the ones associated to (i) the uncertainty in the reference, i.e. the long-term 10-minute rain gauge records, and to (ii) the use of traditional approaches on the long-term rain gauge records (i.e., Generalized Extreme Value distribution fit of the annual maxima series using the method of the L-moments by Hosking et al., 1990).

### 4 Results

### 4.1 Validation of the radar-derived return levels

The methodological uncertainty in extreme return levels estimated using SMEV from the 10-minute rain gauges full record is rather low (FSEs for 100-year return levels are generally <20%, median across stations and durations is ~12%, Fig. 2a, Gauges+SMEV) despite the length of the 10-minute station records ($17.9 \pm 6.6$ years). In contrast, traditional extreme-value methods applied to the full rain gauge records lead to FSE of 20-50% (median across stations and durations ~31%, Fig2a,

Gauges+GEV). This confirms the robustness of the SMEV approach for analyzing these relatively short sub-daily records. When estimating return levels combining weather radar and rain gauges as proposed here, a typical FSE on the 100-year return levels of 15-35% (median across stations and durations ~22%, Fig2a, Radar+SMEV) is revealed. Out of the 65 validation points, only five show FSE exceeding 50% and two exceeding 75% (Fig. 2b; see Fig. S3 for more details). These errors in the radar-derived values are somewhat larger than the uncertainties characterizing our reference, but are lower than the typical

errors obtained using traditional extreme-value approaches on rain gauge data.


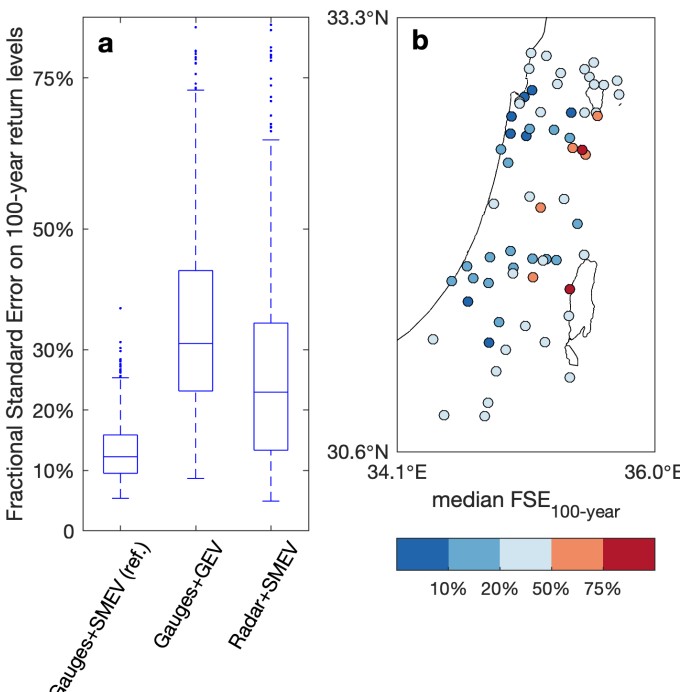

**Figure 2: Validation of the radar-derived extreme return levels. (a) Fractional Standard Error (FSE) associated to the estimated 100-year return levels from i) the reference, i.e. application of SMEV to the gauge full records (Gauges+SMEV), ii) traditional extreme-value approaches on the gauge full records (Gauges+GEV), and iii) radar-derived return levels (Radar+SMEV); all durations are considered. (b) Spatial distribution of the median (across all durations) FSE for the 100-year return levels estimated using SMEV on the radar archive.**

## 4.2 Coastal effect on design precipitation intensities

The impact of the coastline on extreme precipitation statistics is explored using density plots of the parameters of the intensity distributions (Fig. 3) and of return levels (Fig. 4) as a function of the distance from the coastline (see Fig. 1). Of course, this could not be achieved using rain gauges, since there are no systematic rainfall observations over the sea. Over the Mediterranean Sea (negative distances), the scale parameter tends to be lower, and increases while approaching the coast. The offshore dependence of the scale parameter on the distance from the coastline seems rather independent on duration. In the first 20 km inland, the scale parameter seems to decrease for short durations and to weakly increase for long durations. Since the offshore distance from the coast is somehow correlated to the distance from the radar and no rain gauges are available over the sea for the adjustment procedure, one could claim these low offshore values could be caused by radar range effects. To ensure this is not the case, we compared them with the results from a convection-permitting weather model, which is obviously not impacted by the radar observation geometry. The red solid lines superimposed on Fig. 3a-d show the normalized maximum intensities averaged over 41 heavy precipitation events derived from a convection-permitting Weather Research and Forecasting (WRF) model of spatial resolution similar to the radar (1 km$^2$; see details in Armon et al., 2020). The weather model shows the same behavior as the radar, confirming that this is indeed a climatic signal.



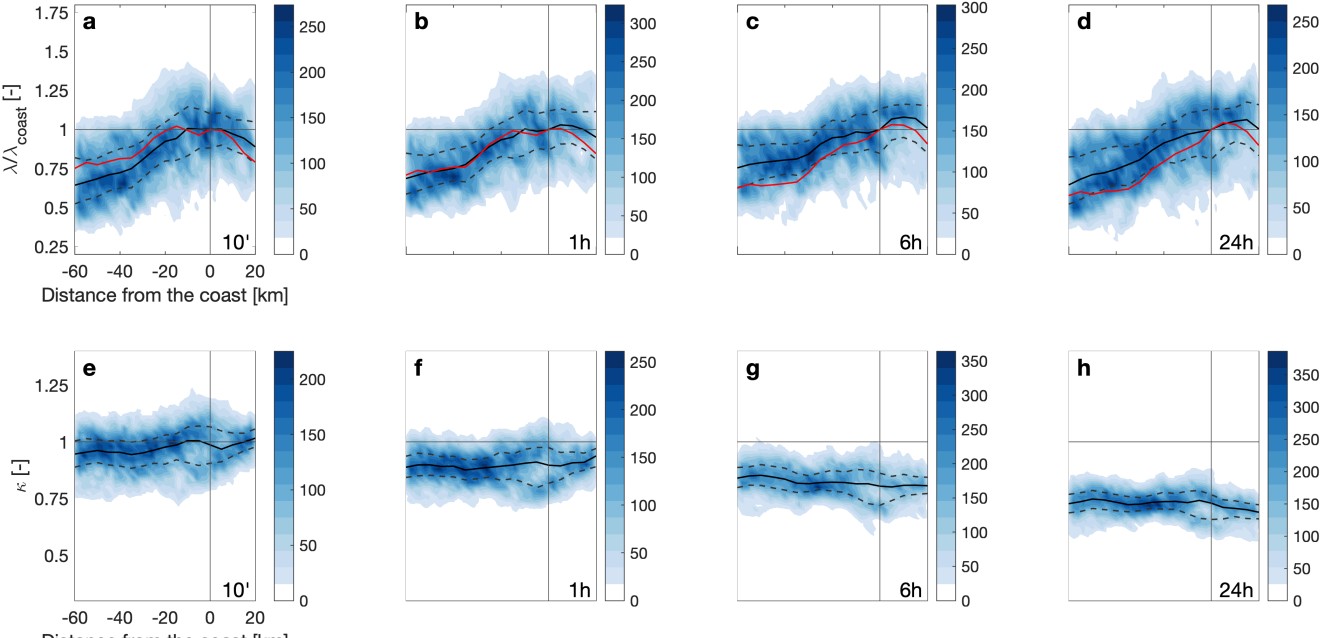

**Figure 3. Density plots of the scale (a-d) and shape parameters (e-h) as a function of the distance from the coastline (see dashed lines in Fig. 1) for durations of 10 minutes, 1 hour, 6 hours and 24 hours. Color shading represents the number of radar pixels in bins of 2 km by 0.05 (a-d) and 2 km by 0.01 (e-h). Negative distances represent offshore areas. The median and 25th-75th quantiles of the distribution along 5-km bins are shown as solid black lines and dashed black lines, respectively. In order to have comparable values across durations, scale parameters (a-d) are normalized over the median value at the coast. For comparison, solid red lines show the median of the normalized maximum intensities from convection-permitting Weather Research and Forecasting (WRF) model simulations of 41 heavy precipitation events in the region (details in Armon et al., 2020). This confirms the decrease offshore is a climatic signal and not an artifact due to distance from the radar and lack of rain gauges for the adjustment (see Sect. 5).**





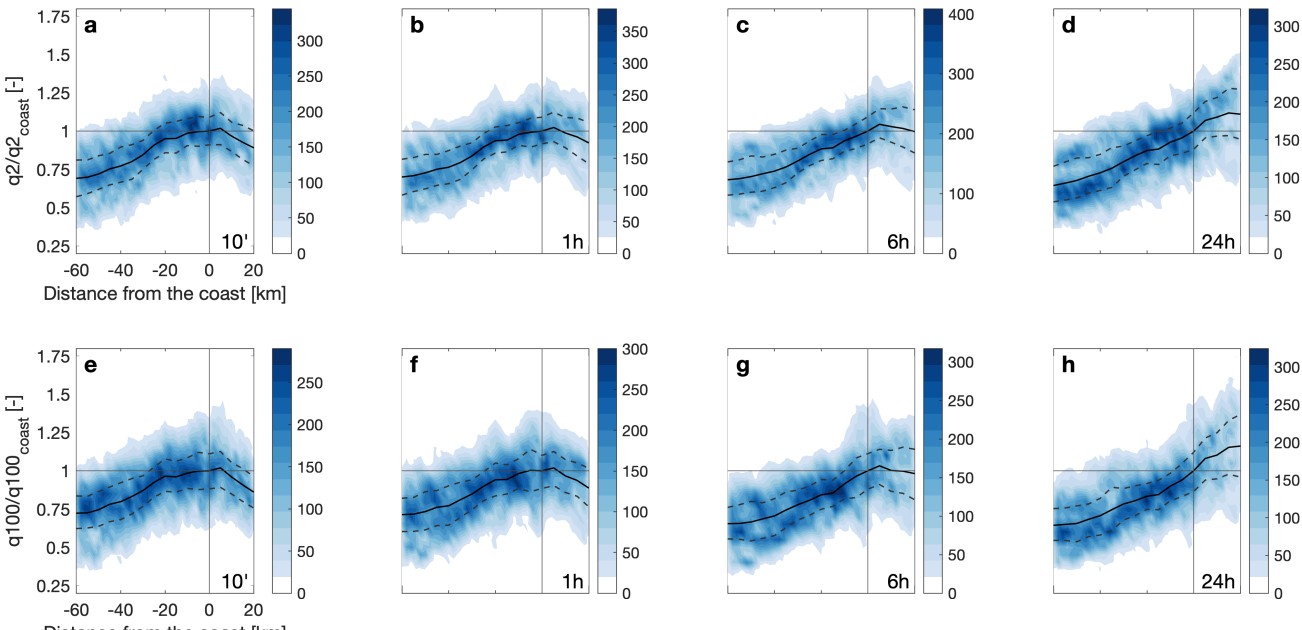

**Figure 4. Density plots of the 2-year (a-d) and 100-year return levels (e-h) as a function of the distance from the coastline (see dashed lines in Fig. 1) for durations of 10 minutes, 1 hour, 6 hours and 24 hours. Color shading represents the number of pixels in bins of 2 km by 0.05. Negative distances represent offshore areas. The median and 25th-75th quantiles of the distribution along 5-km bins are shown as solid black lines and dashed black lines, respectively. In order to have comparable values across durations, return levels are normalized over the median value at the coast.**

No appreciable variation of the shape parameter as a function of the distance from the coastline is observed offshore (Fig. 3e-h), except a mild increase approaching the coast at short durations and a mild decrease at long durations (both not statistically significant). This implies that no significant change in the heaviness of the intensity distribution tail is observed offshore. In general, the shape parameter offshore is slightly lower than 1 at 10-minute duration (50% of the pixels have shape values of 0.8-1) and systematically decreases to ~0.75 at 24-hour duration. Inland, the shape tends to increase at short durations (decreasing tail heaviness) and to decrease at long durations (increasing tail heaviness). The average yearly number of storms (parameter n), presented in Fig. S2 shows no significant coastal pattern.

Figure 4 shows the dependence of 2-year return levels (which correspond to the median annual maximum value) and 100-year return levels (1% yearly exceedance probability) on the distance from the coastline. At all durations, low values are observed over the offshore areas in the Mediterranean Sea, which increase towards the coast. As return levels directly respond to the SMEV parameters and no variations on the shape parameters were found offshore, this is mostly related to the scale parameter patterns highlighted above. Conversely, drastically different effects between short and long durations are observed inland. Extreme return levels of short-duration (up to ~1 hour) tend to peak at or around the coastline, while multi-hour return levels



tend to further increase inland, an effect which is more important for more severe return levels. This comes as a response to the combined increase in both the scale parameter and the tail heaviness (Fig. 3).

## 4.3 Orographic effect on design precipitation intensities

Orography in the area is weakly related to the distance from the radar, and higher terrain may require the need of higher radar sample volumes to avoid ground echoes and blockages. In particular, due to the lack of rain gauge data and to the high elevation of the radar beam above the second orographic barrier, radar estimates of the scale parameter are here expected to be biased. Therefore, the orographic effect on extreme precipitation statistics is here examined focusing on the tail heaviness of the intensity distribution (i.e., the shape parameter), which is not expected to be impacted by systematic biases (see also Sect. 4.4). Density plots of the estimated shape parameters as a function of the local terrain elevation are presented in Fig. 5a-d, together with the corresponding estimates based on the rain gauge full records. The quantitative behavior reported by the radar is consistent with the one from the stations, with significant decreases in tail heaviness at short durations (30 minute and 1 hour) and significant increases at long durations (24 hours) (Fig. 5e). In particular, opposing responses emerge at sub-hourly (increasing tail heaviness with elevation), hourly (decreasing) and daily (increasing) time scales. At very short durations (10-minute), radar estimates show a statistically significant increase of tail heaviness with elevation (negative slope of the regression in Fig. 5e) while rain gauges show a non-significant decrease. This slight difference can be explained by the fact that no rain gauge is available on the rather large and high-elevation Jordanian plateau (east-most portion of the radar domain; Fig. 1). The spatially dense radar sampling increases the statistical significance of the derived slopes, allowing an improved appreciation of these relations. Notably, in the rift valley where terrain elevation is below the sea level, the shape parameter increases with elevation at all durations, although achieving a quantification of the effect is here difficult due to the small size of the region and to the small number of available rain gauges.

The spatial variability of the shape parameter also depends on duration for both rain gauges and weather radar, as shown by the residuals from the regression lines (Fig. 5). The standard deviation of the residuals (Fig. 5f) decreases by ~45% between 10 minutes (~0.09) and 24 hours (~0.05) durations; this decrease seems to be mostly occurring between hourly and daily durations. This implies that larger-scale patterns characterize the statistics of longer-duration extremes and that larger variability could be related to shorter-duration extremes dominated by convective processes.





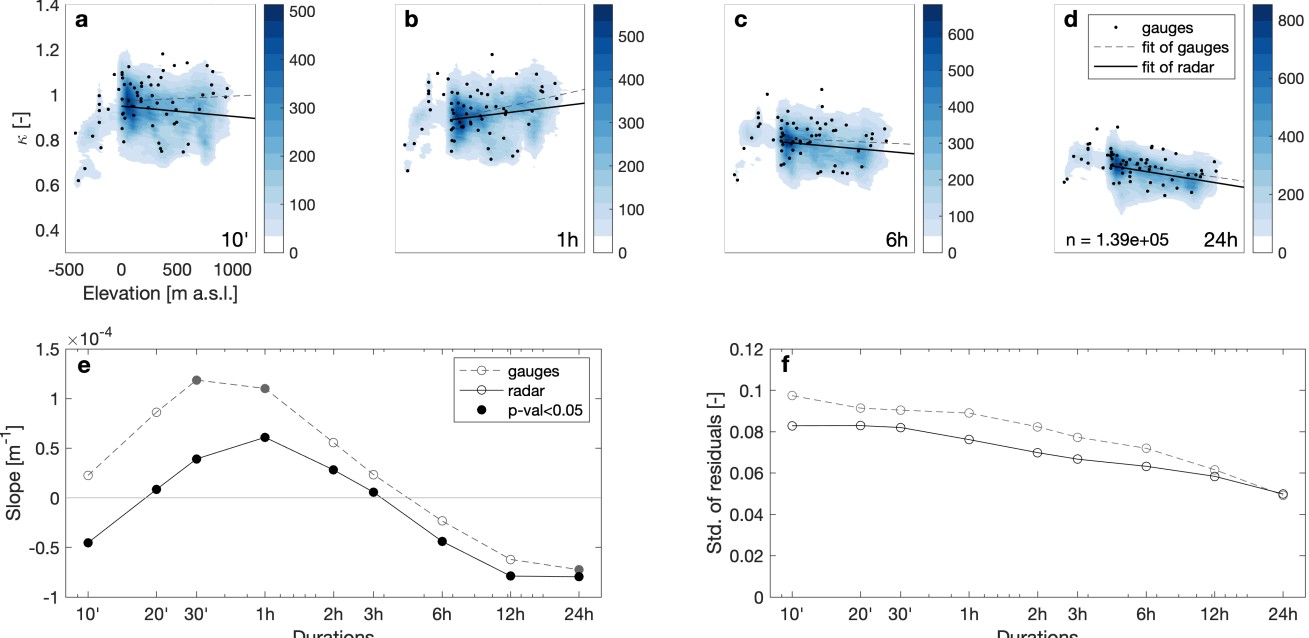

**Figure 5. (a-d) Density plots of the shape parameter (representing the intensity distribution tail heaviness) as a function of terrain**
**elevation for durations of 10 minutes, 1 hour, 6 hours and 24 hours. Color shading represents the number of pixels in bins of 50 m**
**by 0.01. Shape parameters estimated from the 10-minute rain gauges are shown as black dots. Regression lines computed for all the**
**positive elevations (i.e., excluding the rift valley area depression, which shows a different behavior, and including slopes on both the**
**east and west side of the Judean mountains) from weather radar (solid black lines) and 10-minute rain gauges (dashed grey lines)**
**are shown. (e) Slope of the shape-elevation regressions computed from weather radar (solid black lines) and gauges (dashed grey**
**lines) as a function of duration. Statistically significant slopes (p < 0.05) are displayed with filled circles. Negative slopes imply an**
**increase in tail-heaviness with height, and vice versa. (f) Standard deviation of the residuals around the regressions for weather**
**radar (solid black lines) and gauges (dashed grey lines) as a function of duration.**

## 4.4 Intensity distribution parameters and return levels across longitudinal transects

Longitudinal variations of the intensity distribution parameters (Fig. 6) and of return levels (Fig. 7) are further examined
analyzing the three longitudinal transects characterized from west to east by a sea-land boundary, a mountain range, a major
valley and another mountain range. The transects are obtained averaging the values over a 10-km region surrounding the
latitudes (32.4, 31.7 and 31.5°N, see Fig. 1). In order to appreciate variations at multiple durations, scale parameter and return
levels are normalized, at each duration, over the maximum value along the longitudinal direction.

At shorter durations, the scale parameter peaks in the coastal region between few km offshore and 20-30 km inland, and then
decreases, roughly in correspondence of orography (Fig 6a, c, e). Conversely, as duration increases, the peak of the scale
parameter tends to become wider and to shift inland along the orographic ascent to the first orographic barrier. The rift valley
appears as a band with decreased scale parameter, especially at very short and long durations. For hourly and multi-hour



durations the scale parameter seems to increase again around the second orographic barrier (the Jordanian mountains), although
here the high elevation of the radar beam (brown dashed lines) and the absence of 10-minute rain gauges to be used for
adjusting the radar parameters prevent a good quantitative assessment.

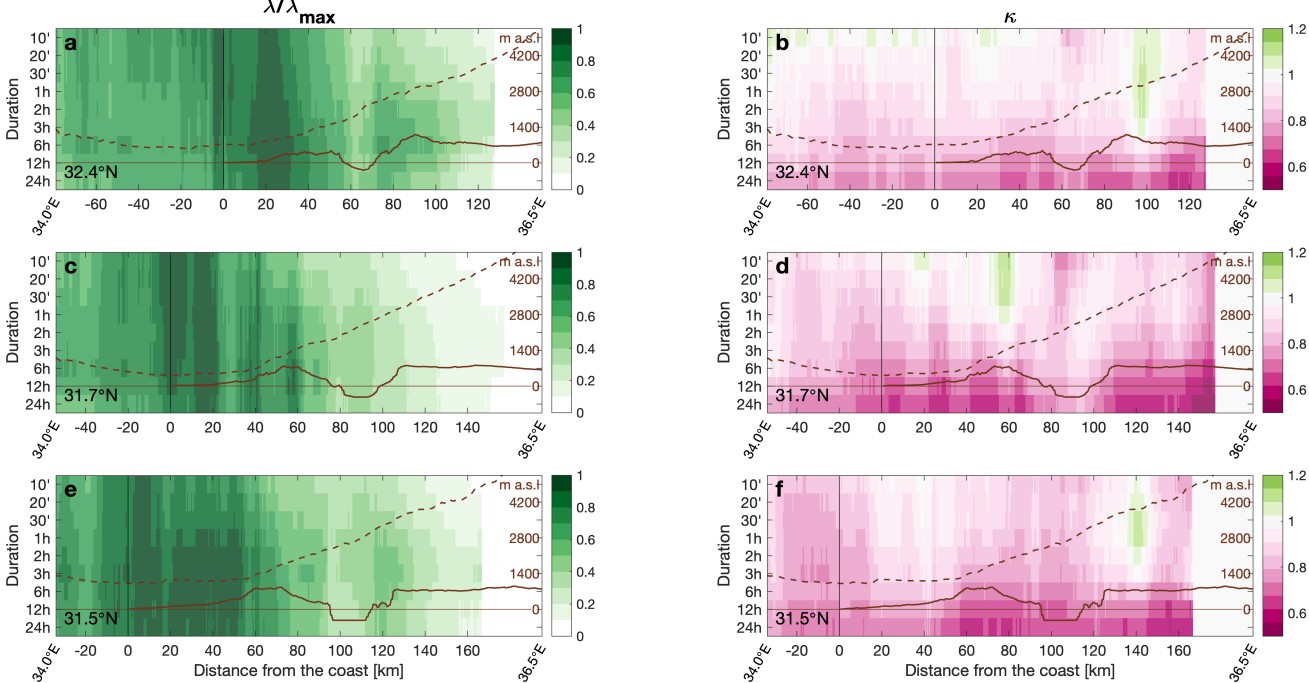

**Figure 6. Longitudinal variations of the scale (a, c, e) and shape (b, d, f) parameters along the three transects (see Fig. 1) as a function
of the distance from the coastline (x-axis) and of duration (y-axis). Transects are obtained averaging the 10-km region surrounding
the three latitudes. The terrain profile and the sampling height of the lowest non-blocked radar beam are superimposed as solid and
dashed lines, respectively (see right hand-side y-axis). In order to have comparable values, the scale parameter (a, c, e) at each
duration is normalized over the maximum value along longitude.**

Overall, the shape parameter tends to decrease with increasing duration, implying an increasing tail heaviness (Fig 6b, d, f).
More specifically, exponential tails (shape parameters close to 1, white colors) and even lighter (higher than 1, green colors)
are observed at 10-minute durations, while shape parameters as low as 0.6-0.7 (heavy tails, pink colors) are typically observed
at 12-24-hour durations, especially in the northern transect. The rift valley is characterized by opposing tail behaviors, with
heavier tails at 10-minute durations and slightly lighter tails at long durations. Orography seems to induce lighter tails at short
durations (≤2 hours), with cases of light tails (green colors) corresponding to the first (Judean mountains, Fig. 6d) and second
(Jordanian plateau, east of the mountaintops, Fig. 6b, f) orographic barriers, although this behavior appears less systematic
from the transects alone.





Extreme return levels directly reflect the spatial variability of the parameters (Fig. 7). The 2-year and 100-year return levels examined here show qualitatively similar behaviors, although quantitative differences exist due to the variability of the tail heaviness (shape parameter, Fig. 6b, d, e) and of average yearly number of storms (Fig. S2). At short durations, return levels

peak within a ~20-40 km strip around the coastline. Increasing the duration, the peak gradually moves inland, generally around the first orographic barrier where the largest daily amounts are found. In all cases, the signature of the rift valley depression at hourly and multi-hour durations is clear, with largely decreased values. However, in response to the sharp increase in tail heaviness, 100-year return levels at sub-hourly duration seem to be higher in the rift than in the surrounding areas. This behavior, already reported in previous studies based on traditional methods (Marra and Morin, 2015; Marra et al., 2017), is

also found in the 2-year return levels over the central transect, where the sharpest orographic gradient between the Judean mountains and the rift valley bottom is found. The impact of the second orographic barrier largely resembles the one of the first orographic barrier (decrease of short duration return levels, increase of long duration return levels), although the numbers are here smaller, at least partially due to the underestimation of the scale parameter discussed above. This impact is most pronounced over the northern transect, where the Jordanian plateau mountaintops are highest, and in all cases is evident over

the upslope part of the mountains.

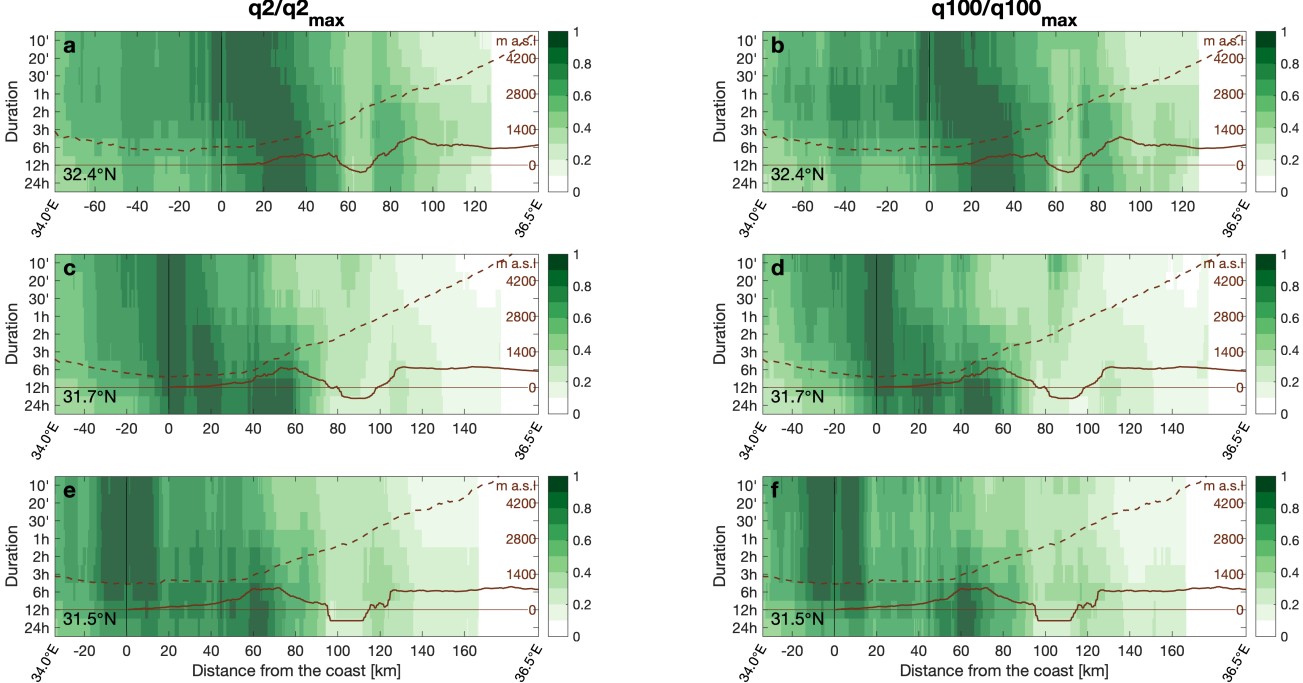

**Figure 7.** Longitudinal variations of the 2-year (a, c, e) and 100-year (b, d, f) return levels along the three transects (see Fig. 1) as a function of the distance from the coastline (x-axis) and of duration (y-axis). Transects are obtained averaging the 10-km region

surrounding the three latitudes. The orographic profile and the sampling height of the lowest non-blocked radar beam are





**superimposed as solid and dashed lines, respectively (see right-hand side y-axis). In order to have comparable values, the return levels at each duration are normalized over the maximum value along longitude.**

## 5 Discussion

The objective of this study is to examine and quantify the relative impacts of coast-land interface and orography on extreme
precipitation statistics and design precipitation intensities. While no direct quantification of the return levels is sought here, the proposed method proved skillful in deriving precipitation intensities as high as the 1 in 100 years event. Estimates of 100-year return levels obtained combining 11 years of weather radar data and 10-minute rain gauges using the here-proposed approach show a median fractional standard error of ~22%. This is lower than the fractional standard error of daily amounts in the weather radar archive (Fig. S1). Combining remotely sensed precipitation estimates with rain gauges using the here-
proposed SMEV framework can thus provide spatially-distributed estimates of extreme return levels which are more accurate than the ones obtained using traditional extreme-value methods directly on point-scale rain gauge records (median fractional standard error ~31%). Nevertheless, given the low methodological uncertainty, the errors in the radar-derived return levels are to be mostly associated to radar errors.

### 5.1 Coastal and orographic effects on design precipitation intensities

A schematic of the observed coastal and orographic effects on extreme precipitation intensities is presented in Fig. 8. Consistently across the study region, we find an increase in tail heaviness with increasing duration. This was previously reported by Marra et al. (2020) and Marra et al. (2021b) and contrasts with previous studies examining the tail of sub-daily intensities (Papalexiou et al., 2018). Nevertheless, as discussed in Marra et al. (2020), the contrast is only apparent and is explained by the different definition of the intensities examined here (ordinary events, that are storm maximal intensities), and
in previous studies (all non-zero intensities, which come with drastically different number of events at different durations). Exponential tails (shape parameters close to 1, white colors in Fig 6b, d, f) and even lighter (higher than 1, green colors) are observed at 10-minute durations, while shape parameters as low as 0.6-0.7 (heavy tails, pink colors) are typically observed at 12-24-hour durations, especially in the northern transect. These observations agree with the theoretical calculations by Wilson and Tuomi (2005) which, under idealized conditions, predict a stretched exponential distribution with shape parameter equal
to 2/3 for extreme storm accumulations (and therefore long-duration intensities). An exception to this behavior is the rift valley, which represent a downwind depression to incoming storms from any direction.

Return levels tend to be much smaller offshore than along the coast (median values 60 km offshore are ~60-75% of the values at the coastline). This comes as a consequence of the decreased scale parameter of the ordinary events distribution. To ensure this is not related to systematic underestimation due to radar range degradation, we compared it with results from convection-
permitting WRF model runs of 41 heavy precipitation events in the area (see Armon et al., 2020), which are obviously not impacted by the radar observation geometry. The weather model (red solid lines in Fig. 3a-d) shows the same behavior as the





radar, confirming that this is a climatic signal. Lower offshore return levels, although affected by large noise, were also found using traditional extreme-value methods with both radar and satellite observations (Marra et al., 2017).

Short duration return levels in the region tend to peak in a 20-40 km band around the coastline and in the rift valley, while long
duration return levels tend to peak around orographic barriers. The regime shift occurs between durations of 1-12 hours, and is more pronounced over the northern portion of the domain (Fig. 6, 7). In contrast, over the southern portion of the domain the effect is mostly observed over durations >6 h, with the coastline exhibiting the largest return levels over durations from 10 minutes to 6 hours (Fig. 6, 7). In general, opposing responses to orography are observed at sub-hourly (increasing tail heaviness with elevation), hourly (decreasing) and daily (increasing) time scales (Fig. 8). The orographic impact on tail heaviness peaks
around hourly durations, as previously reported in Marra et al. (2021b) for the specific case of Mediterranean cyclones. Our results show that this behavior is maintained also when the combined effect of Mediterranean cyclones and other weather systems is considered, although it should be recalled that Mediterranean cyclones account for the majority of storms in most of the examined domain.

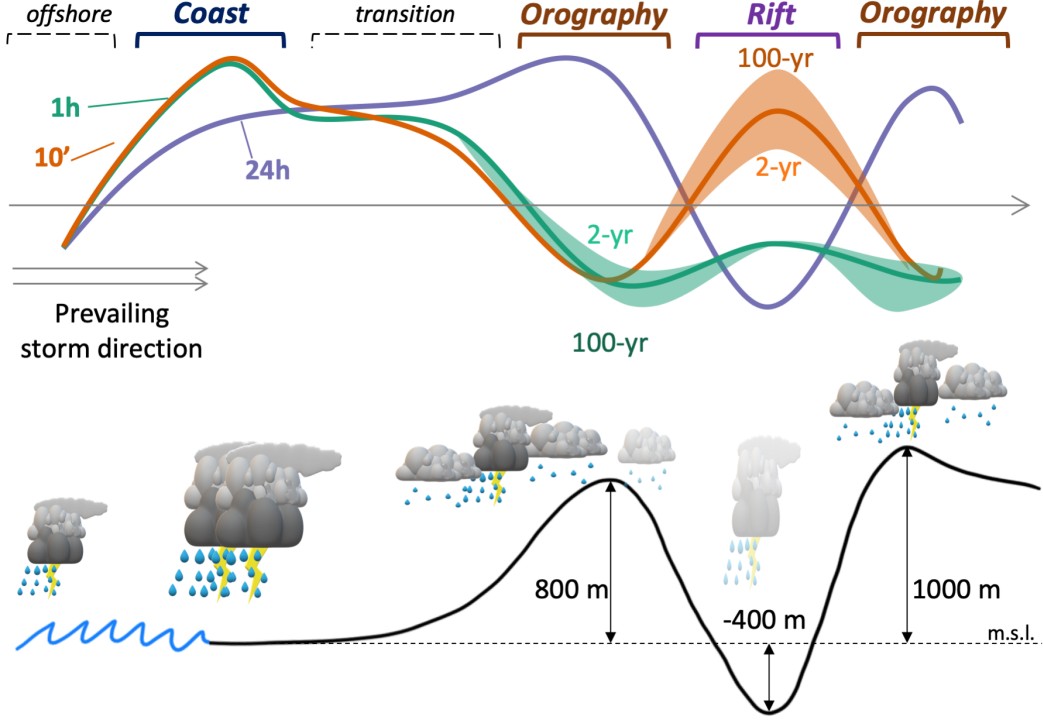


**Figure 8. Schematic representation of the coastal and orographic effect on extreme precipitation. The qualitative impact of local terrain features (sea-land interface, orographic barriers, rift valley depression - displayed on the bottom with a black line) on extreme precipitation return levels of 10 minutes (orange line), 1 hour (green line) and 24 hours (purple line) is summarized. The different response of mild (2-year, i.e. 50% yearly exceedance probability) and extremely severe (100-year, i.e. 1% yearly exceedance**



**probability) return levels of short-duration (10 minutes, 1 hour) to orographic features, which comes as a consequence of the response of the shape parameter of the ordinary events distribution, is displayed using color shades. In the bottom portion of the figure, regions where short-duration convective processes dominate extreme precipitation are indicated with thunderstorms, while regions in which stratiform-like processes and the aggregation of many rain cells dominate extreme precipitation are indicated with thinner precipitating clouds surrounding the thunderstorms. Semi-transparent clouds in the rift indicate that the local occurrence**
**frequency of precipitation events is lower.**

## 5.2 Potential physical mechanisms governing the relation of extreme precipitation with coastline and orography

The shifting of extreme rainfall peak from the coastline towards the mountains with increased duration may be attributed to the different mechanisms involved in the generation of extreme intensity rainfall at different durations. While a detailed meteorological-climatological analysis for the different mechanisms is out of the scope of this paper, there is still a place to
mention possible and known causes for these differences. At very short durations, the main driver of high intensity rainfall is instantaneous rain intensity within convective rain cells. This intensity is affected by the amount of moisture in the air and its ascent rate. While moisture for most rain events in the area arrives from the Mediterranean Sea (e.g., Alpert and Shay-El, 1994; Armon et al., 2019) and would cause higher rain rates over the sea, low level convergence that increases air ascent rates are apparently rather constant over the sea, and increase approaching the coastline. This increase is presumably related to two
main factors: (i) the abrupt increase in drag coefficient at the coast, and (ii) the interaction of mesoscale and synoptic winds in presence of a curved coastline (e.g., Colle et al., 2002). The increase in drag coefficient, caused by larger roughness of the land versus the sea, slows down landward winds and triggers a low-level velocity convergence (e.g., Bergeron, 1949; Roeloffzen et al., 1986; Colle and Yuter, 2007). Neglecting friction, one can commonly think of large-scale atmospheric horizontal wind flows as a balance between the Coriolis, the centrifugal and the pressure gradient forces. This approximation, called the
gradient flow, is violated close to the ground, where friction plays a larger role (chapter 4 in Holton, 2004). When increased friction is added upon a "balanced" gradient flow, such as in proximity of the coastline, a secondary circulation regime is initiated, in which winds in a cyclonic flow tend to flow inward towards the low-level center (chapter 5 in Holton, 2004). This change in wind direction may increase wind curvature and thus increase low-level air ascent. Similarly, the interaction of meso-scale winds such as land-sea breeze with the synoptic wind, especially in the presence of a curved coastline, may enhance
instantaneous rain rates because of meso-scale convergence between the two sources of wind (e.g., Neumann, 1951; Andersson, and Gustafsson, 1994; Brummer et al., 1995).

Examination of the time of the day in which the highest short-duration intensities (i.e., the ordinary events in the distribution tail, defined here as the largest 45%) are observed (Fig. 9) shows that the highest offshore intensities tend to occur in the early morning (0:00-6:00 local time) or morning (6:00-12:00) and they shift to mostly morning (6:00-12:00) at the coastline and
near inland. This supports the above proposed mechanism, in which the convergence created by the superposition of the westerly winds typical of Mediterranean cyclones with land breeze, which is expected to peak in the early morning hours when the sea is the warmest compared to the land, could be a contributing factor for the short-duration intensity peak around the





coastline. High short-duration intensities in regions where orographic lifting typically dominates (the west side of the Judean mountains and the Jordanian plateau) tend to peak in the afternoon (12:00-18:00), suggesting that the diurnal-thermal

instability driven by the heating of the ground, could here interact with orographic lifting enhancing the convective vertical motions. Conversely, on the lee-side of the Judean mountains, short-duration intensities seem to peak in the morning hours (6:00-9:00), although this signal is not confirmed by the in-situ rain gauges (Fig. 9), which show peaks more concentrated in the early afternoon (12:00-15:00). This mismatch might be caused by the large elevation difference between the radar sampling volumes (here ~3000 m a.s.l. and above, see profiles in Fig. 5 and Fig. 6) and the terrain elevation where rain gauges measure

(even 400 m below the mean sea level): the radar might systematically overshoot the core of shallow convection and, in addition, the stronger convection captured by the radar might be overestimated due to evaporation of the hydrometeors along their falling path.

Farther away from the radar, in both the eastern and southern areas where climate is drier (Fig. 1), short-duration afternoon-to-night peaks are prevailing (north, east, and south of the Dead Sea). This may reflect the higher impact of local thunderstorms,

which rely on the diurnal heating to bring air ascent to high-enough elevations independent of orography (Sharon and Kutiel, 1986; Dayan et al., 2001; Armon et al., 2019; Dayan et al., 2021). This might explain the reversed behavior with regions affected by orography, as well as the heavier tails previously observed in these arid areas (Marra et al., 2015; Marra et al., 2017).

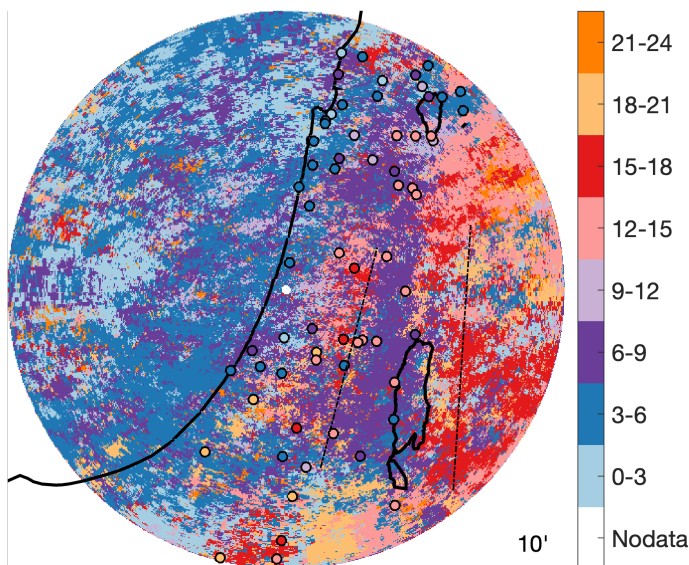


**Figure 9. Time of the day in which the highest 45% of the ordinary events (i.e., the storm maximal intensities defining the tail of the intensity distribution) at 10-minute duration are typically observed. Circles colored with the same color scheme show the corresponding values obtained from the 10-minute rain gauges. Dot-dashed lines show the approximate location of the peaks of the two main orographic barriers.**




In contrast to short duration rainfall, high multi-hour intensities in the region are mainly a result of the aggregation of multiple rain cells over a specific location and to the shift of extremes towards more stratiform-like processes. They are thus less dependent on the instantaneous rain rates, and are expected to increase in regions where spatial and temporal rainfall coherence is enhanced. The mountain range causes extreme rain cells to get surrounded by regions with lower-intensity stratiform-like

precipitation, and to cover a larger area with lower rain rates (Peleg and Morin, 2012; Marra et al, 2021b). Additionally, the orographic lifting tends to squeeze more efficiently the remaining moisture from the air to provide more rainfall in total (Houze et al., 2001; Roe, 2005). Indeed, previous studies in the region found that the area of rain cells is increased inland (Karklinski and Morin, 2006; Peleg and Morin, 2012) and that the temporal autocorrelation of rainfall is larger over the mountains compared to the coastal region (Marra et al, 2021b).

Figure 10 presents for each pixel the duration at which the relative rank (among all pixels) of its 2-year and 100-year return level is the highest. Short duration (10-20 minute) intensities dominate offshore and inland in proximity of the shoreline, over the rift valley, and in the southern portion of the domain, while long durations (6-24 hours) dominate in the inland areas, where orography is more pronounced. This pattern resembles the results seen previously (e.g., Fig. 8), highlighting the governing hydrometeorological patterns. Namely, short-duration high-intensity convective processes are the main reason for heavy

precipitation close to the coastline where moisture is plentiful, which makes the short duration events (orange colors) being ranked first in this region. On the other side, longer duration processes require the climbing of air parcels over the mountains (e.g., Alpert, 1986; Roe 2005), which causes a reorganization of the convective processes (with differential responses at 10-minute and 1-hour scales), and leads to orographic rainfall, with long-lasting stratiform-like processes. This causes mountainous regions to be characterized by long duration extremes ranking first (purple regions in Fig. 10). Descending from

the mountaintops (Fig. 1 and Fig. 9) air parcels dry and precipitation is inhibited. In these regions, where conditions are more arid, and along the desert at the southern part of the region, rainfall is spottier and more local (e.g., Sharon and Kutiel, 1986), and its extremes are generally related to rare but sometimes violent convection, which is translated into orange colors in Fig 10. In the north of the region, where valleys connect the Dead Sea rift with the Mediterranean (see Fig. 1), the water divide is lower in elevation, and extremes tend to peak at intermediate durations (~3-6 hours, light green and light purple colors in Fig.

10). Similar patterns are also observed in the northern shoreline, where mountain reliefs are close to the coastline, possibly due to the simultaneous impact of coastal and orographic effects.





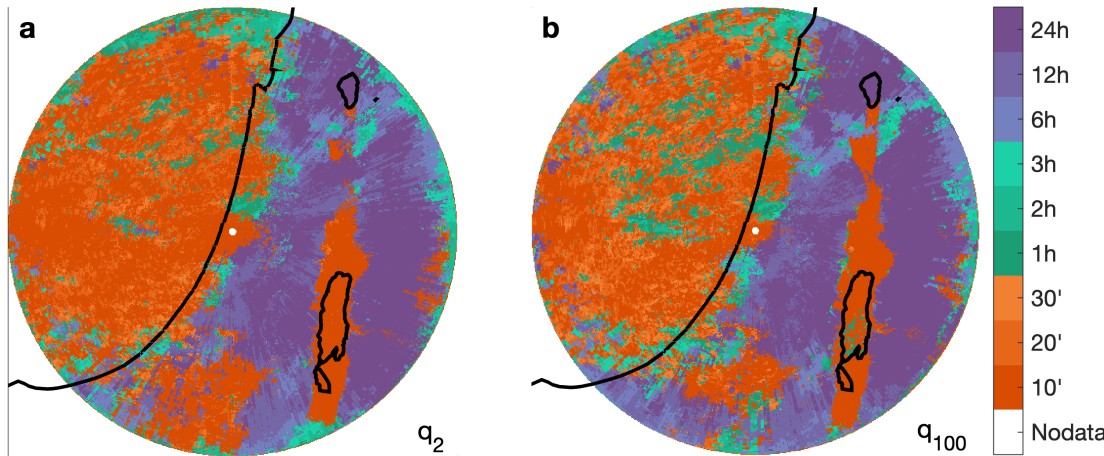

**Figure 10. Durations at which the local 2-year (a) and 100-year (b) return levels are the strongest: at each duration, intensities are ranked among all pixels, so to associate 1 to the largest value over the region and 0 to the smallest; the maps display the duration which, at each pixel, has the largest rank.**

### 5.3 Implications for risk management

Roughly 50% of the population in the region is concentrated in the coastal plains, mainly in cities within 10-20 km from the coastline (CBS, 2016; Bitan and Zviely, 2019). Population growth and migration are expected to disproportionally increase the population density in these coastal cities (Neumann et al., 2015; UN-DESA 2018). Short-duration precipitation intensities are crucial triggers of pluvial urban flooding, which cause casualties and important losses every year (Cristiano et al., 2017; Villarini et al., 2010). Our results, made possible by the use of distributed rainfall estimates from a weather radar, show that these intensities are stronger in the highly-populated coastal areas (orange colors in Fig. 10) of the region, with important implications for urban flood risk management. Moreover, future projections show an increase in the short duration extreme precipitation intensities in the area (Armon et al., 2021). Given the alarming projections in terms of sea-level rise, explicit consideration of the compound impact of extreme rain intensities and storm surges under future scenarios should be specifically addressed in future planning of coastal cities (Zscheischler et al., 2018).

Considering flash flood generation and watershed response, it is interesting to note that short durations (10-20 minutes, orange colors in Fig. 8 and Fig. 10) intensities tend to dominate extremes offshore, over the coastal regions and in the rift, while longer durations (6-24 hours, purple colors in Fig. 8 and Fig. 10) tend to dominate extremes inland and over the orographic regions. These spatial patters coincide in some cases with the characteristic response time of watersheds. As mentioned above, short durations extremes can lead to urban flooding over the vast built area near the shore, but also in arid watersheds in the rift and in the south of the region (see areas with mean annual rainfall < 200 mm yr-1, Fig. 1), due to the typical fast hydrological response of these areas (e.g., Morin et al., 2001; Zoccatelli et al., 2019). Conversely, watersheds draining inland regions north



and west of the Judean mountains are typically covered by agricultural areas, forests or other natural land uses and would

typically show few-hours characteristic time scales (Zoccatelli et al., 2019); for example, time of concentration of watersheds
with ~50 km length and ~600 m height (numbers in the range of the watersheds west of the Judean mountains) is ~3.5 hours
based on Johnstone and Cross (1949).

**6 Summary and conclusions**

We propose and test a novel methodology for extreme precipitation frequency analysis that allows to adjust relatively-short

archives of weather radar precipitation estimates using rain gauges as a reference. The approach is based on the simplified
Metastatistical extreme value (SMEV) formulation (Marani et al., 2015; Marra et al., 2019; Marra et al., 2020) and relies on
the adjustment of the remotely sensed information in the SMEV parameter-space. The methodology provides spatially-
distributed, high-resolution statistical descriptions of extremes which accurately reflect the statistics of in-situ observational
records. Combining 11 years of weather radar data with 10-minute rain gauge data in the southeastern Mediterranean, we

obtain estimates of 100-year return levels characterized by standard errors in the order of ~22%, which is lower than the
standard errors obtained using traditional approaches on rain gauge data. We then used the obtained spatially-distributed, high-
resolution statistical descriptions of extremes to describe and quantify the distinct signature of coastlines and orography on
extreme precipitation statistics and design precipitation intensities at multiple sub-daily durations.

The SMEV formulation allows to examine the statistics of extremes based on two main parameters, a scale parameter, which

describes the scale of the whole intensity distribution, and a shape parameter, which describes the tail heaviness, that is the
proportion between extreme and mild events. The scale parameter of the rain intensity distributions tends to be lower offshore
and to increase approaching the coast at all durations (10 min to 24 hours). At short and hourly durations, the scale parameter
peaks at the coastline, while at longer durations it tends to peak a few kilometers inland corresponding to the orographic
barriers. The distributions tail heaviness is rather uniform above the sea and rapidly changes inland, with opposing directions

at short (decreasing tail heaviness) and long (increasing tail heaviness) durations. The orographic impact on the intensity
distributions shows a decrease of tail heaviness at short durations, which peaks at hourly durations, and an increase in tail
heaviness at long durations; this confirms previous results based on the systematic interaction of Mediterranean cyclones with
a regular orographic barrier (Marra et al., 2021b), and suggests this could be a more general behavior not strictly related to the
physical processes typical of Mediterranean cyclones in the region. The overall impact on extreme return levels results in

increasing return levels moving offshore towards the coastline and in opposing behaviors at different durations inland, with
decreasing extremes at short and hourly durations and increasing extremes at long durations. Short- and hourly-duration return
levels tend to peak in proximity of the coastline; long-duration return levels tend to peak inland corresponding to orographic
ascents and mountaintops (Fig. 8).

We identify three main regimes which respond differently to coastal and orographic forcing: (i) very short durations (~10

minutes), mainly related to the peak intensity of convective rain cells (intensity-dominated regime); (ii) hourly durations,



related to the rainfall yield (and thus to the spatiotemporal organization) of individual convective cells; (iii) long durations (~6-24 hours), related to the accumulation of multiple convective cells as well as to long-lasting stratiform processes (Fig. 8). Convective processes are accounted for higher extremes near the coastline; this is presumably related to increased drag coefficient and to interactions of the synoptic and mesoscale systems, as suggested by the early-morning timing of coastal
extremes. Conversely, the short-duration extremes in orography-dominated regions tend to peak in the afternoon, suggesting a contribution from the diurnal thermal instability (Fig. 9). Long-duration extremes are higher in regions where spatial and temporal rainfall coherence is enhanced, such as orographic regions where orographic lifting smooths the precipitation field. The distinct coastal and orographic effects at different durations are expected to modulate the temporal scales of the most relevant local impacts (Fig. 10): short-scale hazards (e.g., urban floods, debris flows) are more of concern for the coastal and
rift regions, while longer-scale hazards (e.g., flash floods) could be more relevant over mountainous areas.

Given the relatively flexible requirements in terms of availability and record length of weather radar and rain gauge data, the here presented methodology is deemed suitable for extreme multi-duration precipitation frequency analyses in many other regions worldwide. Specifically, it could help (i) improving the quantification of local return levels without the need for homogeneity and/or temporal scaling assumptions and (ii) increasing the understanding of the local climatology of extreme
multi-duration precipitation. In addition, (iii) it could be combined with information from climate models and local constraints (e.g., Marra et al., 2021a) to derive observationally-sound projections of future extreme return levels of interest for climate change impact studies.

**Acknowledgements**

This study was funded by the Israel Science Foundation (grant no. 1069/18), is a contribution to the HyMeX program, and is
based upon work from COST Action CA19109 "MedCyclones" supported by COST - European Cooperation in Science and Technology (www.cost.eu). FM thanks the Institute of Atmospheric Sciences and Climate (ISAC) of the National Research Council of Italy (CNR) for the support. The authors thank Uri Dayan for the fruitful discussions.

**Author contributions**

Conceptualization: FM, MA, EM; Formal analyses: FM, MA; Data curation: FM, MA; Paper writing: FM; Paper review and
editing: FM, MA, EM

**Data availability**

Rain gauge data were provided and preprocessed by the Israel Meteorological Service and freely available at: https://ims.data.gov.il/ (Hebrew only). Original weather radar data was provided by the Israel Meteorological Service





**Competing interests**

The authors declare no conflict of interests.

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
