# Peer review of "Coastal and orographic effects on extreme precipitation revealed by weather radar observations"

_Hydrology and Earth System Sciences, 2021_

## Referee Comment (RC1)

Investigation of extreme hydro-meteorological events in complex terrain, for example, the coastal and orographic areas, attracts increasing attention especially under the climate change. However, the interaction effect of weather system and terrain on extreme precipitation remains unclear. The manuscript entitled "Coastal and orographic effects on extreme precipitation revealed by weather radar observations" provides deep insights into the understanding of the interaction effect of weather system and terrain features on local extreme precipitation using radar rainfall data. It was high-quality from the experiment design to the effect analysis and discussion, as well as the excellent language expression. Nevertheless, I would like to point out several key questions and suggestion for the authors.

(1) In Abstract, the expression of "we obtain estimates of the 1 in 100 years intensities" was obscure. Did it mean the precipitation intensities?

(2) In Line 146 of Page 6, please list the mathematic equation of FSE index with detailed explanation of variables and parameters.

(3) How was the SMEV model constructed and applied to different extreme precipitation data? In Section 3.1, I highly advice the authors to use the mathematic equations to express the SMEV model structure, exceedance probability, and return levels. Math language is more precise than the text description. In addition, please introduce the novelty of SMEV.

(4) Were the storm and ordinary events defined only based on rain gauge data, or separately defined using rain gauge and radar extreme precipitation estimates? Furthermore, the storm events were individually extracted using multiple extreme precipitation datasets with various durations, is it right? Why the parameter $n$ is the same for all durations? Please the author make it clear.

(5) About parameter $n$, how to use it in SMEV model in the Steps 3 and 4 of Page 7? Also, if using mathematic formula, it is easy to clarify the unnecessary confusion. Meanwhile, please make it italic here and hereinafter.

(6) In Line 217 of Page 8, it's doubtful that the authors implemented the bias correction and spatial interpolation of radar extreme precipitation (steps 3 to 5) based on SMEV parameters rather than precipitation itself. For the multiple parameter

optimization problem, there exists "parameter equifinality" phenomenon. Namely, very different parameter sets may lead to similar result (referring to probability distribution in this study). Therefore, a numerical value nearby the optimal parameter may be an unavailable one. Maybe we cannot "correct" or "interpolate" the estimated parameters derived from SEVE model. This is very important to the whole study. Please ensure it testable, refer and list several typical previous studies with this usage.

(7) The expression of "intensity distribution" occurs frequently throughout the manuscript. I know it meaning "precipitation intensity distribution" (as Line 49). However, to be accurate, I suggest the authors use "precipitation intensity distribution (PID)" or "probability density function (PDF)" instead of "intensity distribution".

(8) In Line 255, the GEV approach and its full name (generalized extreme value distribution) should be presented in Section 3 for the method.

(9) In Figure 3 (a-d), what does the proportion of the scale parameter represent? For the subplots (e-h) of shape parameter, there is no benchmark line in red, why?

(10)  In Figure 5, only subplot (b) of 1 h duration displayed an increasing trend for the shape parameter with increasing elevation. However, the scatters and color shading in Fig. 5 (b) were very similar with those in Fig.5 (a). Please the authors recheck and discuss this inconsistency in trend.

---

## Author Comment (AC1)

**Response to reviewer #1**

Investigation of extreme hydro-meteorological events in complex terrain, for example, the coastal and orographic areas, attracts increasing attention especially under the climate change. However, the interaction effect of weather system and terrain on extreme precipitation remains unclear. The manuscript entitled "Coastal and orographic effects on extreme precipitation revealed by weather radar observations" provides deep insights into the understanding of the interaction effect of weather system and terrain features on local extreme precipitation using radar rainfall data. It was high-quality from the experiment design to the effect analysis and discussion, as well as the excellent language expression. Nevertheless, I would like to point out several key questions and suggestion for the authors.

We would like to thank the reviewer for the time taken to read our manuscript and for the constructive comments. We are sure they will help us better delivering our results in our revised manuscript.

We would also like to point out here that, following an improvement in the codes, some of the figures will be slightly updated in the revised version of the manuscript. These changes are minimal and do not affect the results nor the conclusions of the study.

1. In Abstract, the expression of "we obtain estimates of the 1 in 100 years intensities" was obscure. Did it mean the precipitation intensities?

Yes, here the meaning is as the reviewer suggests. To improve clarity, we will edit the text to *"we obtain estimates of the 1 in 100 years precipitation intensities"*.

2. In Line 146 of Page 6, please list the mathematic equation of FSE index with detailed explanation of variables and parameters.

The definition of these metrics comes down to the correlation coefficient and the root mean square error, with which all readers of this journal are surely familiar. We think that including the full equations would make the text heavy for no practical reason, so we would suggest not to address this specific comment. But we remain open to further discussion on this.

3. How was the SMEV model constructed and applied to different extreme precipitation data? In Section 3.1, I highly advice the authors to use the mathematic equations to express the SMEV model structure, exceedance probability, and return levels. Math language is more precise than the text description. In addition, please introduce the novelty of SMEV.

Equations behind the SMEV approach are already present in mathematical form in the manuscript: the Weibull distribution is shown in line 170, the SMEV distribution in line 202. For improved clarity, we will introduce its use since the beginning of the section: *"We use the framework for sub-daily precipitation frequency analysis based on the concept of ordinary events presented in Marra et al. (2020), and we rely on the simplified Metastatistical extreme value (SMEV) framework (Marra et al., 2019) for the estimation of extreme return levels"*.
We will also include some additional background on the method: *"This approach is equivalent to ordinary statistics of n-sized block maxima under independence for which the tail behavior is known (Serinaldi et al., 2020)."*
Finally, we will add some information to improve the presentation: (i) we will use the suffix $D$ to the parameters $\lambda$ and $\kappa$, which depend on the duration $d$, in the equations - thus making them $\lambda_D$, $\kappa_D$, and (ii) we will specify once more that $n$ is independent of $D$.

4. Were the storm and ordinary events defined only based on rain gauge data, or separately defined using rain gauge and radar extreme precipitation estimates? Furthermore, the storm events were individually

extracted using multiple extreme precipitation datasets with various durations, is it right? Why the parameter *n* is the same for all durations? Please the author make it clear.

Thank you for pointing out these aspects, which we deem crucial. Storms are uniquely defined using the rain gauge data, as detailed in 3.1 point 1, and are, by definition, always the same for all durations. This is a key property of our statistical model, as shown in Marra et al, 2020 and underlined in the manuscript (3.2 point 2).
First, storms are defined at the regional scale, uniquely for stations and radar, based on station data, as detailed in point 1 of section 3.1, while ordinary events are defined at each rain gauge or radar pixel. We will improve clarity on this by explicitly saying *"at each station/pixel"* where relevant, and by editing the sentence in 3.1 point 1 to: *"For both rain gauge and radar analyses, storms are defined…"*; and we will add explicitly that *n* is the same at all durations in 3.1 point 3: *"the parameter n [...] is thus the same at all durations"*. Also, as detailed in the response to the comment above (comment #3), the SMEV equation will explicitly shows the suffix *D* in the scale and shape parameters (which depend on duration) and will not use the suffix for *n*.

5.  About parameter *n*, how to use it in SMEV model in the Steps 3 and 4 of Page 7? Also, if using mathematic formula, it is easy to clarify the unnecessary confusion. Meanwhile, please make it italic here and hereinafter.

We will use italics for the parameter. Its use was already detailed in the mathematical expressions (section 3.1, point 4), so we don't see a way to further address this comment, if not by specifying the definition of *n* also right after it is used: *"...where ζ is the sought yearly exceedance provability (e.g. 1% for the 100-year events), n the average yearly number of storms, and…"*.
As said above, we will also improve the information on 3.1 to improve the overall clarity on this point.

6.  In Line 217 of Page 8, it's doubtful that the authors implemented the bias correction and spatial interpolation of radar extreme precipitation (steps 3 to 5) based on SMEV parameters rather than precipitation itself. For the multiple parameter optimization problem, there exists "parameter equifinality" phenomenon. Namely, very different parameter sets may lead to similar result (referring to probability distribution in this study). Therefore, a numerical value nearby the optimal parameter may be an unavailable one. Maybe we cannot "correct" or "interpolate" the estimated parameters derived from SEVE model. This is very important to the whole study. Please ensure it testable, refer and list several typical previous studies with this usage.

We thank the reviewer for stressing this important aspect. While in principle "equifinality" can be considered, since we estimate the parameters of the Weibull distribution using a minimization (least square regression in transformed coordinates), we respectfully disagree on its potential impact on our analysis:
First, it should be noted that, although our model has 3 parameters, one of them (*n*) is defined prior to the parameter estimation, as it depends on the definition of the storms. Consequently, only two parameters potentially contribute to this problem, thus decreasing the potential impact.
Second, a strict validation procedure (50% of the stations are removed and used for validation) has been applied and presented, showing that the radar-based SMEV is more accurate than traditional methods based on stations alone. If equifinality was a significant problem for our application, the validation would have been not satisfactory.
Nevertheless, while working on this comment, we realized should be made clearer: the SMEV model is not linear, and the adjustment we apply in the parameter space should be considered an approximation as soon as we move away from the rain gauge locations. This approximation will be good close to the stations and its accuracy will tend to decrease in the most ungauged regions. In this sense, however, it should be recalled that the error metrics we provide in the validation are representative of the model accuracy and include errors due to this approximation. To clarify this aspect, we will include a clearer caveat sentence in the methods:
*"Since the SMEV model is not linear, the adjustment we apply in the parameter space is to be considered an approximation, whose accuracy potentially decrease with increasing distance from the stations."*
As suggested by the reviewer, a possible alternative to this approach could be to adjust directly the quantiles. Albeit useful for design applications, we think this would not be helping our target, which is of examining

the coastal and orographic impacts on the statistical properties of the storms (hence the parameters), in addition to the quantiles. Moreover, it would require new computations every time a new return level is sought.

7. The expression of "intensity distribution" occurs frequently throughout the manuscript. I know it meaning "precipitation intensity distribution" (as Line 49). However, to be accurate, I suggest the authors use "precipitation intensity distribution (PID)" or "probability density function (PDF)" instead of "intensity distribution".

In this study "intensity distribution" refers to the cumulative distribution function, rather than the probability density function. For consistency with recent works on the topic and to avoid confusion with the extreme value cumulative distribution function, we would prefer to keep the use of "intensity distribution", but we will define this clearly at its first use in section 3.1: *"Once the cumulative distribution function describing their tail, here termed "intensity distribution", is known…"*

8. In Line 255, the GEV approach and its full name (generalized extreme value distribution) should be presented in Section 3 for the method.

Thank you for pointing this out. We will introduce the acronym at the end of section 3.3, where the method is presented.

9. In Figure 3 (a-d), what does the proportion of the scale parameter represent? For the subplots (e-h) of shape parameter, there is no benchmark line in red, why?

The figure caption explains what this normalization represents: *"In order to have comparable values across durations, scale parameters (a-d) are normalized over the median value at the coast."* (lines 287-288 of the submitted manuscript). We will also include a new figure in the supporting information in which the non-normalized variables are presented.
Weather model runs in the area only cover 41 individual storms so far, and are thus not yet sufficient to compute the parameters of the intensity distribution and of the SMEV model. Nevertheless, the median of the normalized maximum intensities across these storms represents a reasonable proxy of the spatial variability of scale parameter of this distribution. Basically, the red lines for the scale parameter are the most we can extract from these convection-permitting model runs. We will edit the text to make this aspect clear: *"…solid red lines show the median of the normalized maximum intensities (which can be considered as a proxy for the scale parameter) from…"*

10. In Figure 5, only subplot (b) of 1 h duration displayed an increasing trend for the shape parameter with increasing elevation. However, the scatters and color shading in Fig. 5 (b) were very similar with those in Fig.5 (a). Please the authors recheck and discuss this inconsistency in trend.

We believe the reviewer refers to the trends in solid black (derived from radar data) but compares them with the black points (gauge data). Indeed, gauge data showed positive trend for Fig. 5 (a) (although not significant), as shown in Fig. 5 (e). However, we cannot agree on the fact that the color shading in (a) and (b) are similar and should thus present the same trend. Please note that, following an improvement in the codes we run, this figure will be slightly updated in the revised version of the manuscript.

---

## Author Comment (AC2)

**Response to reviewer #2**

The authors study the effects of land-coast interactions and orography over a complex study area on extreme precipitation. The work reveals how short (radar) time series could be used to look at several features of a study area in depth. They use the return levels themselves as well as the underlying parameters to study these effects over different durations ranging between 10-minutes and 1-day.

This is a high-quality manuscript, that is well-structured, well-written, and contains a lot of detailed information while still conveying the main message. Therefore, this review only consists of some minor points and clarifications:

We would like to thank the reviewer for the time taken to read our manuscript and for the constructive comments. We are sure they will help us better delivering our results in our revised manuscript.

We would also like to point out here that, following an improvement in the codes, some of the figures will be slightly updated in the revised version of the manuscript. These changes are minimal and do not affect the results nor the conclusions of the study.

1. Figure 1: The lines with the annual rainfall amounts are hard to distinguish from the underlying elevation in the mountainous areas. Perhaps the authors could change the colors, or add an extra panel containing the annual rainfall amounts. Also add some more information on the transects, as it only becomes clear much later on why these transects are included.

Thank you for the suggestion. We will update the figure to address the request and we will edit the caption to better specify the meaning of the transects: *"…location of the three transects shown in Fig. 6 and Fig. 7"*.

2. L147-151: It would be helpful for the readers if the authors add some information on which ranges of the FSE are considered good, and how much this "large improvement over the previous radar archive available for the region" is.

It is rather subjective to quantify what is "good" in terms of these metrics, especially for the FSE. What we can do is to include a quantification of these quantities together with a direct comparison with the previous radar archive for the area. We will update section 2.3 to include this information: *"These metrics imply that the root mean square error of the daily amounts is 107% of the average rain gauge daily amount in wet days (≥0.1 mm), and that ~42% of the variance is to be imputable to measurement errors. These results represent a large improvement over the previous radar archive available for the region, which had correlation coefficient below 0.5 and FSE between 1.5 and 2 (Marra and Morin, 2015, personal communication). These improvements are attributable…"*. It should be noted that the metrics in Marra & Morin 2015 were presented for hourly and yearly scale, while here the daily scale is reported (hence the "personal communication").

3. L151-L159: what are the implications of the issues of the radar that still remain? Which issues generally cause over or underestimation, or in which regions are the results likely over/underestimated?

We will improve this paragraph to provide some additional details: *"Despite the evaluation metrics, some issues still remain in the final estimates, which could decrease the quantitative accuracy of the estimated return levels: [...] Underestimation due to range effects is also visible in the northern and southern portions of the domain for areas farther than ~100 km from the radar"*.

4. Section 3.1 point 1 (L181-L188): what are these 2 weather types? Are they two of the ones introduced in the study area? Why do they need to be separated by 1-day dry periods?

We will improve the clarity of this aspect: *"Storms are defined at the regional scale as consecutive wet periods of the same weather type separated by 1-day dry periods or by the change to a different weather type. [...] Weather types are defined according to the classification presented in Marra et al. (2021a), which*

*is a simplification of the semi-objective classification by Alpert et al. (2004) and classifies wet days into two types: (i) Mediterranean cyclones, (ii) other types (mostly active Red Sea troughs, see Sec. 2.1).".* As stated in the manuscript, separating or not separating them will not affect the results of this study and, strictly speaking, they don't need to be separated for the analyses developed in this study; however, in order to have compatibility with future planned studies in which the separation is necessary, we keep this separation alive also in this study: *"It is worth noting that, although the framework allows to separately consider different weather types, the use of a unique weather type is not crucial for the accurate estimation of return levels in the study area (see Marra et al. 2019); separate consideration of the 2 weather types is here included to grant compatibility with future studies involving climate projections, for which the distinct use of weather types is crucial (e.g., Marra et al., 2021a)."*

5. Section 3.3: Make the part of using GEV for comparison more prominent, and provide the abbreviation in this section already. The abbreviation a few lines further now comes without an introduction.

Thank you for pointing this out. We will introduce the acronym at the end of section 3.3, where the method is presented, and provide a better explanation of the underlying methodological details in-text rather than in parenthesis: *"Traditional methods are here constructed using the block-maxima approach: series of annual maximum values are extracted from the full rain gauge records; parameters of the fitting Generalized Extreme Value (GEV) distribution are estimated using the method of the L-moments (Hosking et al., 1990)."*. Overall, this method is well-known, and in the paper GEV is only used in Fig. 2a; we hope this modification will suffice our needs.

6. L258: change to: "only seven show FSE exceeding 50% of which two exceeding 75% (Fig. 2b; see Fig. S3 for more details on other durations)".

Please note that, following an improvement in the codes we run, this figure will be slightly updated in the revised version of the manuscript. We will update the text to: *"...only nine show FSE exceeding 50%, of which three exceed 75%..."*

7. Figure 3: add ticks on the x-axes for 3e-h. Would it work for such density plots to have 1 colorbar representative of all sub-panels for easier comparison?

We will add the ticks to Fig 3 and also to Fig. 5 which shared the same visualization issue, thanks for pointing it out. Unfortunately having 1 colobar for all the panels would make them extremely difficult to read; it was our initial choice but we eventually realized that having multiple colorbars is better.

8. Section 4.4: why are these the longitudinal transects chosen over these 3 latitudes?
   Consider introducing this in the method section, possibly around Figure 1 where they are just mentioned in the caption.

We will improve the caption in Fig. 1 to provide additional information on the transects: *"...location of the three transects shown in Fig. 6 and Fig. 7..."*. We will also provide details about the choice of these specific transects, but we feel it would be more appropriate to do so in section 4.4: *"The location of these transects is chosen based on radar visibility and on the presence of regular orographic profiles."*

9. Figures 6 and 7 are normalized, which does provide interesting information and helps the reader in understanding the differences along transects or orography. However, it would be interesting to also include some actual values, for instance of the T2 and T100 estimates, also over different durations.

Thank you for the suggestion, we will provide a figure analogous to Fig. 6 (only scale parameter) and Fig. 7 without normalization in the supporting information as Fig. S4.

10. L372: Do you mean middle transect instead of northern?

Indeed, thank you for pointing this out, we will update the text accordingly.

11. L372-L373: The patterns of the rift valley described aren't visible in 3f, consider adding: "for the northern two transects".

We will remove this sentence from the paper.

12. Figure 9: Consider changing using a circular colormap as this one is hard to interpret.

Thanks for the suggestion, we will update the colormap as recommended.

---

## Referee Report (RR1)

The revised manuscript entitled "Coastal and orographic effects on extreme precipitation revealed by weather radar observations" has been improved. Most of the comments and questions were fully responded in detail. This study is high-quality for the publication in HESS. Nevertheless, several remaining and new questions still exist and are listed as follows:

(1) I don't think a statement of FSE and/or a citation of it would take much space in the text, since this definition, which was used throughout the result, is basic and important for the method.

(2) Please mind the line numbers in your reply should refer to the revised manuscript.

(3) The word of "framework" was used in Method Section for many times, for examples, in Lines 168, 169 and 228. However, sometimes the so-called "framework" was just a statistical distribution or a simple utility.

(4) Some subscripts of the variables were italic, while some are not (see Lines 235 and 245). Please correct them throughout the manuscript.

(5) In supplement, the color change trend in FSE legend of Fig. S1a (dark colors denote large values) is inconsistent with that in BIAS legend of Fig. S1b (light colors denote large values). Furthermore, this trend in FSE legend of Fig. S3 changes again.

(6) The $\lambda$ scale parameter in Fig. S4a, d, g has a unit of mm/h. Why a distribution parameter has precipitation intensity unit. Please clarify its physical meaning.

(7) In Fig. 5a, for the shape parameter as a function of terrain elevation for durations of 10 minutes, the weather radar data derived a slightly decreasing regression line (in solid black), whereas the rain gauge data derived a slightly increasing one (in dashed grey). It can be observed from the first two dots at 10' in Fig. 5 also. What results in this discrepancy, the radar errors or the bias correct uncertainty?

(8) Thanks for the authors' reply to my Comment 4 about the storm definition. For the radar pixels on the sea (I mean the regions where there are no rain gauges, see the Fig. S2), what station data or which rain gauges were referred when the storm events of radar precipitation were extracted?

(9) The Section 6 Summary and Conclusion should be more concise and without citations, since detailed analysis has been provided in the Section 5 Discussion.

(10) In Line 618, I believe the authors made a mistake by writing "standard errors in the order of ~22%," when revising the manuscript. Please recheck all the updated numerical values after every revision.

---

## Editor Decision (ED1)

Hydrol. Earth Syst. Sci. Discuss., referee comment RC1
https://doi.org/10.5194/hess-2021-395-RC1, 2021

[Figure]

**Comment on hess-2021-395**

Anonymous Referee #1

Referee comment on "Coastal and orographic effects on extreme precipitation revealed by weather radar observations" by Francesco Marra et al., Hydrol. Earth Syst. Sci. Discuss., https://doi.org/10.5194/hess-2021-395-RC1, 2021

Investigation of extreme hydro-meteorological events in complex terrain, for example, the coastal and orographic areas, attracts increasing attention especially under the climate change. However, the interaction effect of weather system and terrain on extreme precipitation remains unclear. The manuscript entitled "Coastal and orographic effects on extreme precipitation revealed by weather radar observations" provides deep insights into the understanding of the interaction effect of weather system and terrain features on local extreme precipitation using radar rainfall data. It was high-quality from the experiment design to the effect analysis and discussion, as well as the excellent language expression. Nevertheless, I would like to point out several key questions and suggestion for the authors.

- In Abstract, the expression of "we obtain estimates of the 1 in 100 years intensities" was obscure. Did it mean the precipitation intensities?
- In Line 146 of Page 6, please list the mathematic equation of FSE index with detailed explanation of variables and parameters.
- How was the SMEV model constructed and applied to different extreme precipitation data? In Section 3.1, I highly advice the authors to use the mathematic equations to express the SMEV model structure, exceedance probability, and return levels. Math language is more precise than the text description. In addition, please introduce the novelty of SMEV.
- Were the storm and ordinary events defined only based on rain gauge data, or separately defined using rain gauge and radar extreme precipitation estimates? Furthermore, the storm events were individually extracted using multiple extreme precipitation datasets with various durations, is it right? Why the parameter $n$ is the same for all durations? Please the author make it clear.
- About parameter $n$, how to use it in SMEV model in the Steps 3 and 4 of Page 7? Also, if using mathematic formula, it is easy to clarify the unnecessary confusion. Meanwhile, please make it italic here and hereinafter.
- In Line 217 of Page 8, it's doubtful that the authors implemented the bias correction and spatial interpolation of radar extreme precipitation (steps 3 to 5) based on SMEV parameters rather than precipitation itself. For the multiple parameter optimization

problem, there exists "parameter equifinality" phenomenon. Namely, very different parameter sets may lead to similar result (referring to probability distribution in this study). Therefore, a numerical value nearby the optimal parameter may be an unavailable one. Maybe we cannot "correct" or "interpolate" the estimated parameters derived from SEVE model. This is very important to the whole study. Please ensure it testable, refer and list several typical previous studies with this usage.

- The expression of "intensity distribution" occurs frequently throughout the manuscript. I know it meaning "precipitation intensity distribution" (as Line 49). However, to be accurate, I suggest the authors use "precipitation intensity distribution (PID)" or "probability density function (PDF)" instead of "intensity distribution".
- In Line 255, the GEV approach and its full name (generalized extreme value distribution) should be presented in Section 3 for the method.
- In Figure 3 (a-d), what does the proportion of the scale parameter represent? For the subplots (e-h) of shape parameter, there is no benchmark line in red, why?
- In Figure 5, only subplot (b) of 1 h duration displayed an increasing trend for the shape parameter with increasing elevation. However, the scatters and color shading in Fig. 5 (b) were very similar with those in Fig.5 (a). Please the authors recheck and discuss this inconsistency in trend.

Please also note the supplement to this comment:
https://hess.copernicus.org/preprints/hess-2021-395/hess-2021-395-RC1-supplement.pdf

[Figure]

Hydrol. Earth Syst. Sci. Discuss., referee comment RC2
https://doi.org/10.5194/hess-2021-395-RC2, 2021
**Comment on hess-2021-395**

Anonymous Referee #2

Referee comment on "Coastal and orographic effects on extreme precipitation revealed by weather radar observations" by Francesco Marra et al., Hydrol. Earth Syst. Sci. Discuss., https://doi.org/10.5194/hess-2021-395-RC2, 2021

The authors study the effects of land-coast interactions and orography over a complex study area on extreme precipitation. The work reveals how short (radar) time series could be used to look at several features of a study area in depth. They use the return levels themselves as well as the underlying parameters to study these effects over different durations ranging between 10-minutes and 1-day.

This is a high-quality manuscript, that is well-structured, well-written, and contains a lot of detailed information while still conveying the main message. Therefore, this review only consists of some minor points and clarifications:

- Figure 1: The lines with the annual rainfall amounts are hard to distinguish from the underlying elevation in the mountainous areas. Perhaps the authors could change the colors, or add an extra panel containing the annual rainfall amounts. Also add some more information on the transects, as it only becomes clear much later on why these transects are included.
- L147-151: It would be helpful for the readers if the authors add some information on which ranges of the FSE are considered good, and how much this "large improvement over the previous radar archive available for the region" is.
- L151-L159: what are the implications of the issues of the radar that still remain? Which issues generally cause over or underestimation, or in which regions are the results likely over/underestimated?
- Section 3.1 point 1 (L181-L188): what are these 2 weather types? Are they two of the ones introduced in the study area? Why do they need to be separated by 1-day dry periods?
- Section 3.3: Make the part of using GEV for comparison more prominent, and provide the abbreviation in this section already. The abbreviation a few lines further now comes without an introduction.
- L258: change to: "only seven show FSE exceeding 50% of which two exceeding 75% (Fig. 2b; see Fig. S3 for more details on other durations)".
- Figure 3: add ticks on the x-axes for 3e-h. Would it work for such density plots to have 1 colorbar representative of all sub-panels for easier comparison?
- Section 4.4: why are these the longitudinal transects chosen over these 3 latitudes?

Consider introducing this in the method section, possibly around Figure 1 where they are just mentioned in the caption.

- Figures 6 and 7 are normalized, which does provide interesting information and helps the reader in understanding the differences along transects or orography. However, it would be interesting to also include some actual values, for instance of the T2 and T100 estimates, also over different durations.
- L372: Do you mean middle transect instead of northern?
- L372-L373: The patterns of the rift valley described aren't visible in 3f, consider adding: "for the northern two transects".
- Figure 9: Consider changing using a circular colormap as this one is hard to interpret.

---

## Author Response (AR2)

**Response to reviewer's comments**

Dear authors,
Reviewer #2 gave some suggestions on technical corrections. Please have a look and make corresponding corrections. Thanks for your great efforts to revise the manuscript.
Best regards,
Editor
Thank you for handling our revised manuscript. We provide below here a point-by-point response to the residual comments from reviewer #2. Reviewer's comments are in black font, our reply in blue. We addressed all the reviewer's comments.

The revised manuscript entitled "Coastal and orographic effects on extreme precipitation revealed by weather radar observations" has been improved. Most of the comments and questions were fully responded in detail. This study is high-quality for the publication in HESS. Nevertheless, several remaining and new questions still exist and are listed as follows:
We would like to thank the reviewer for the time taken to read our manuscript and for the comments.

(1) I don't think a statement of FSE and/or a citation of it would take much space in the text, since this definition, which was used throughout the result, is basic and important for the method.
We included the equation for FSE, as follows (lines 147-148): *"(FSE, that is the root mean square error normalized over the average rain gauge amount, computed as: $FSE = \frac{\sqrt{1/N \sum_i (r_i - g_i)^2}}{1/N \sum_i g_i}$, where $g_{i=1...N}$ are the rain gauge observations and $r_{i=1...N}$ the radar estimates for the corresponding pixel, as in Marra and Morin, 2015)"*

(2) Please mind the line numbers in your reply should refer to the revised manuscript.
We are sorry for this inconvenience.

(3) The word of "framework" was used in Method Section for many times, for examples, in Lines 168, 169 and 228. However, sometimes the so-called "framework" was just a statistical distribution or a simple utility.
Thank you for pointing this out. Indeed, we found few instances in which the term was misused. We replaced it with 'method' in line 169, and removed the term in lines 183 and 250.

(4) Some subscripts of the variables were italic, while some are not (see Lines 235 and 245). Please correct them throughout the manuscript.
Thank you for pointing this out, we corrected where relevant.

(5) In supplement, the color change trend in FSE legend of Fig. S1a (dark colors denote large values) is inconsistent with that in BIAS legend of Fig. S1b (light colors denote large values). Furthermore, this trend in FSE legend of Fig. S3 changes again.
It should be noted that the quantities in Fig S1a, S3 have different properties than the ones in Fig. S1b. Fig. S1a and S3 (which have consistent colormap) show FSE (i.e. standard errors), for which low values are good and high values are not. Conversely, Fig. S1b shows multiplicative Bias, for which the perfect score is in the middle and both low and high values are not good; here, we use a diverging colormap, with colors typically used to identify dry/wet, and hence facilitate the reading.
What the reviewer comments is not actually correct, since both the colormaps become darker for worse performance. By chance, underestimation biases in Fig. S1b are more frequent than overestimation biases, so that it may look like lower values in Fig. S1b are darker, but darkness-wise the colormap is perfectly symmetric.

(6) The scale parameter in Fig. S4a, d, g has a unit of mm/h. Why a distribution parameter has precipitation intensity unit. Please clarify its physical meaning.
This comes from the definition of scale parameter: a parameter $s$ is a scale parameter for $F$ when the distribution of $\frac{x}{s}$ does not depend on $s$, i.e. when $F(x, s) = F(\frac{1}{x}, 1)$. Following this general definition, a scale parameter always has the units of the variable at hand.

(7) In Fig. 5a, for the shape parameter as a function of terrain elevation for durations of 10 minutes, the weather radar data derived a slightly decreasing regression line (in solid black), whereas the rain gauge data derived a slightly increasing one (in dashed grey). It can be observed from the first two dots at 10' in Fig. 5 also. What results in this discrepancy, the radar errors or the bias correct uncertainty?

Technically, there is no discrepancy: as shown in Fig. 5e, the slope of the shape-elevation regression from rain gauges in Fig. 5a is not significant, meaning that although it visually seems positive, we cannot claim it is different from zero. Given the adjustment we provide to the radar data, radar-estimated parameters corresponding to the rain gauge locations are the same as the rain gauge parameters. If one uses radar estimates in place of rain gauges and samples only the rain gauge locations (as gauges do), would get the same values and regression line.

Any difference in these regressions is thus to be associate to the fact that radar and gauges sample the area differently. Since rain gauges are small in number, the slopes derived from rain gauges are strongly dependent on the abundance of stations and on their location. The apparent mismatch between radar and gauges in Fig. 5a is due to exactly this problem, and highlights the added value of our results.

Rain gauges in high elevations are few and are systematically located west of the Jordan rift valley (there is no sub-hourly rain gauge data available for the Jordanian Plateau) while the weather radar allows us to sample the whole region.

(8) Thanks for the authors' reply to my Comment 4 about the storm definition. For the radar pixels on the sea (I mean the regions where there are no rain gauges, see the Fig. S2), what station data or which rain gauges were referred when the storm events of radar precipitation were extracted?

As mentioned in our response and in the manuscript, storms are defined at the regional scale as consecutive wet days (section 3.1 point 1). This also holds for the pixels in regions where no gauges are available (either the sea or the Jordanian Plateau). Given the small range we analyze (140 km from the radar) and the climatology of the region (well defined storms separated by relatively long dry periods), this definition is deemed sufficient to also cover the ungauged areas. It should be recalled that, in case no rain is observed in a pixel during a storm, that storm is not considered as a storm for the statistics of that particular pixel.

(9) The Section 6 Summary and Conclusion should be more concise and without citations, since detailed analysis has been provided in the Section 5 Discussion.

Thanks for the suggestion. We shortened the section removing some text and some references. Since our discussion is rather detailed, however, we prefer to keep the main messages in the conclusions, in order to have them delivered to readers with less time available.

(10) In Line 618, I believe the authors made a mistake by writing "standard errors in the order of ~22%," when revising the manuscript. Please recheck all the updated numerical values after every revision.

Good call, this was a leftover from the previous version. We updated it to 26%.